# SLC25A1 and ACLY maintain cytosolic acetyl-CoA and regulate ferroptosis susceptibility via FSP1 acetylation

Wei Li[1,6], Jing Han[2,6], Bin Huang [1,6], Tengteng Xu[1,3,6], Yihong Wan [1,4], Dan Luo[1], Weiyao Kong[1], Ying Yu[1,5], Lei Zhang[1,3,4], Yong Nian [5✉], Bo Chu [2✉] & Chengqian Yin [1,3✉]

## Abstract

**Ferroptosis, an iron-dependent form of programmed cell death characterized by excessive lipid hydroperoxides accumulation, emerges as a promising target in cancer therapy. Among the solute carrier (SLC) superfamily, the cystine/glutamate transporter system antiporter components SLC3A2 and SLC7A11 are known to regulate ferroptosis by facilitating cystine import for ferroptosis inhibition. However, the contribution of additional SLC superfamily members to ferroptosis remains poorly understood. Here, we use a targeted CRISPR-Cas9 screen of the SLC superfamily to identify SLC25A1 as a critical ferroptosis regulator in human cancer cells. SLC25A1 drives citrate export from the mitochondria to the cytosol, where it fuels acetyl-CoA synthesis by ATP citrate lyase (ACLY). This acetyl-CoA supply sustains FSP1 acetylation and prevents its degradation by the proteasome via K29-linked ubiquitin chains. K168 is the primary site of FSP1 acetylation and deacetylation by KAT2B and HDAC3, respectively. Pharmacological inhibition of SLC25A1 and ACLY significantly enhances cancer cell susceptibility to ferroptosis both in vitro and in vivo. Targeting the SLC25A1-ACLY axis is therefore a potential therapeutic strategy for ferroptosis-targeted cancer intervention.**

**Keywords** Ferroptosis; SLC25A1; ACLY; FSP1; Acetylation
**Subject Categories** Cancer; Metabolism; Post-translational Modifications & Proteolysis

## Introduction

Ferroptosis, an iron-dependent form of programmed cell death characterized by the excessive accumulation of lipid hydroperoxides (Dixon et al, 2012; Liang et al, 2022), has emerged as a pivotal pathway in various pathophysiological conditions, such as cancer, ischemia and reperfusion injury (IRI), stroke, and neurodegenerative disorders (Jiang et al, 2021; Yao et al, 2021). Cancer cells have intricate antioxidant defense systems to neutralize lipid peroxidation and evade ferroptosis, including the cyst(e)ine-GSH-GPX4 (Yang et al, 2014), FSP1-CoQH2/VKH2 (Bersuker et al, 2019; Doll et al, 2019; Mishima et al, 2022), DHODH-CoQH2 (Mao et al, 2021), GCH1-BH4 (Kraft et al, 2020; Soula et al, 2020) pathways and most recently reported the 7-DHC system (Freitas et al, 2024; Li et al, 2024) and the sex hormone-dependent MBOAT1/2 system (Liang et al, 2023), each with distinct subcellular localizations to detoxify lipid hydroperoxides and protect against ferroptosis. Accumulating studies have revealed that ferroptosis is implicated in tumor suppression, cancer immunotherapy and drug resistance, suggesting that induction of ferroptosis could represent a new and effective anti-cancer strategy (Chen et al, 2021a; Lei et al, 2022).

FSP1 functions in parallel to GPX4 to mitigate lipid peroxidation through its NAD(P)H-dependent oxidoreductase activity, which reduces CoQ10 to ubiquinone (CoQH2) or vitamin K to VKH2 (Li et al, 2023), rendering tumors resistant to ferroptosis and representing a promising target to trigger cancer cell death. Numerous studies have demonstrated that post-translational modifications play important roles in the regulation of FSP1 functions. N-myristoylation of 2-glycine in FSP1 mediates its plasma membrane recruitment, where the protein functions to suppress ferroptosis (Doll et al, 2019). In addition, the E3 ubiquitin ligases and TRIM69 (Yuan et al, 2022) regulate FSP1 stability through ubiquitination and subsequent proteasomal degradation, while TRIM21-mediated FSP1 ubiquitination at residues K322 and K366 is involved in its membrane localization (Gong et al, 2023). Nonetheless, other mechanisms underlying the regulation of FSP1 function and their impacts on ferroptosis sensitivity and tumor growth remain to be elucidated.

The mitochondrial citrate carrier SLC25A1, or citrate/isocitrate carrier (CIC), is a member of the solute carrier (SLC) superfamily located in the inner mitochondrial membrane (Palmieri, 2013). It is the sole transporter mediating citrate or isocitrate exchange from mitochondria to the cytoplasm against malate (Peng et al, 2020).

[1]Institute of Cancer Research, Shenzhen Bay Laboratory, Shenzhen, Guangdong 518107, China. [2]Department of Cell Biology, School of Basic Medical Sciences, Cheeloo College of Medicine, Shandong University, Jinan, Shandong 250012, China. [3]Shenzhen Medical Academy of Research and Translation (SMART), Shenzhen, Guangdong 518107, China. [4]School of Chemical Biology and Biotechnology, Peking University Shenzhen Graduate School, Shenzhen, Guangdong 518055, China. [5]College of Pharmacy, Nanjing University of Chinese Medicine, Nanjing, Jiangsu 210023, China. [6]These authors contributed equally: Wei Li, Jing Han, Bin Huang, Tengteng Xu. ✉E-mail: ynian@njucm.edu.cn; chubo123@sdu.edu.cn; yincq@szbl.ac.cn

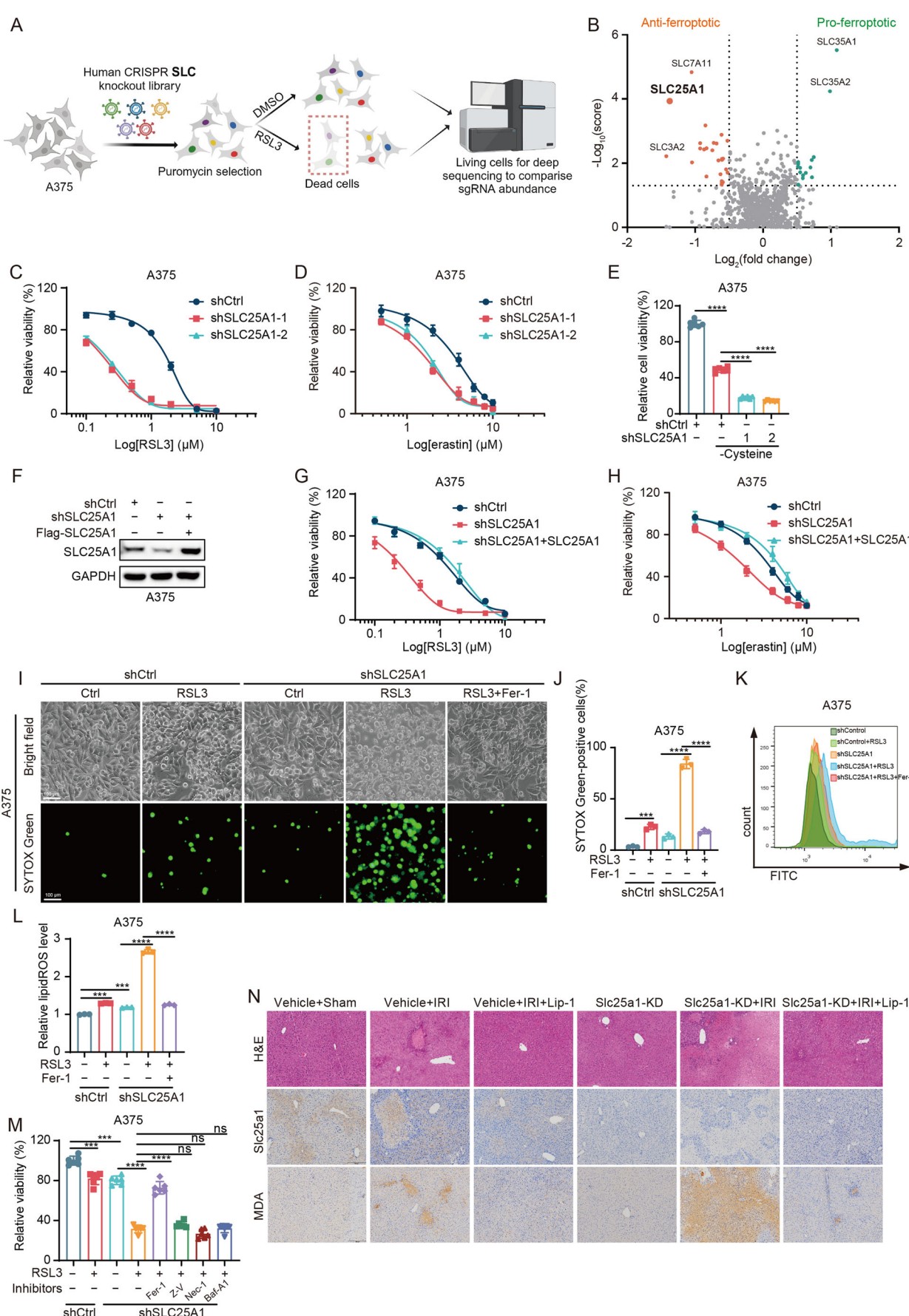

◄

**Figure 1.    Identification of SLC25A1 as a critical regulator of ferroptosis sensitivity.**

(A) Schematic diagram of SLC family-targeted CRISPR-Cas9 screening in A375 cells. (B) Representation of gene scores from CRISPR-Cas9 screening, categorized as anti-ferroptosis and pro-ferroptosis. SLC25A1 is marked in bold. (C–E) Measurement of cell viability in shControl (shCtrl) versus shSLC25A1-A375 cells following treatment with increasing doses of RSL3 (C) or erastin (D) over 48 h, or culture in cysteine-depleted medium for 24 h (E). $n = 3$ for (C, D), and $n = 6$ for (E), $n$ represents biological independent experiments. (E) Statistical analysis by were calculated using one-way ANOVA tests; mean + SD, $p$ values from left to right: ****$p = 1.46E-10$, ****$p = 4.55E-10$, ****$p = 5.56E-11$. (F) Immunoblot of SLC25A1 protein levels in indicated cell groups. SLC25A1-knockdown A375 cells were reintroduced with a shRNA-resistant SLC25A1 construct (shSLC25A1 + SLC25A1-A375). (G, H) Evaluation of cell viability in shCtrl, shSLC25A1, and shSLC25A1 + SLC25A1-A375 cells after 48 h of treatment with increasing concentrations of RSL3 (G) or erastin (H). $n = 3$ for (G, H), $n$ represents biological independent experiments. (I, J) Detection of cell death using SYTOX Green in shCtrl and shSLC25A1-A375 cells treated as indicated for 48 h. RSL3 at 0.5 μM, Fer-1 at 1 μM. Upper panel: phase-contrast images, lower panel: SYTOX Green staining for dead cells. Scale bars: left, 100 μm. (I) Quantitative analysis of cell death percentage is presented. $n = 3$, $n$ represents biological independent experiments. Statistical analysis by two-way ANOVA tests; mean + SD, $p$ values from left to right: ***$p = 0.0002$, ****$p = 1.97E-05$, ****$p = 2.17E-05$ (J). (K, L) Detection of lipid peroxidation through flow cytometer in shCtrl and shSLC25A1-A375 cells under indicated treatments for 6 h. RSL3 at 0.5 μM, Fer-1 at 1 μM (K). Quantification of lipid peroxidation percentage is presented. $n = 3$, $n$ represents biological independent experiments. Statistical analysis by two-way ANOVA tests; mean + SD, $p$ values from left to right: ***$p = 0.0001$, ***$p = 0.001$, ****$p = 1.59E-06$, ****$p = 2.26E-06$ (L). (M) Measurement of cell viability in shCtrl and shSLC25A1-A375 cells subjected to indicated treatments for 48 h. Treatments include RSL3 at 0.5 μM, Fer-1 at 1 μM, Z-VAD-FMK (Z-V) at 10 μM, Necrostatin-1 (Nec-1) at 1 μM, and Bafilomycin A1 (Baf-A1) at 50 nM. $n = 6$, $n$ represents biological independent experiments. Statistical analysis by two-way ANOVA tests; mean + SD, $p$ values from left to right: ***$p = 0.0001$, ***$p = 0.0002$, ****$p = 1.40E-09$, ****$p = 7.84E-08$, ns: not significant. (N) Liver-specific Slc25a1 knockdown (Slc25a1-KD) mice were generated by tail vein injection of AAV8 virus three weeks later. Lip-1 (10 mg/kg) was injected half an hour before ischemia reperfusion (IRI) or sham-treatment. Visual inspection showing liver damage and representative images showing H&E staining, MDA and Fsp1 immunohistochemical staining of livers from mice under the indicated treatment conditions. Scale bars: right, 200 nm. The experiment was repeated at least three times. Source data are available online for this figure.

Beyond its metabolic roles in the TCA cycle, SLC25A1 is essential for de novo lipogenesis and protein acetylation through the regulation of cytoplasmic acetyl-CoA levels (Zhang et al, 2023). Aberrant SLC25A1 activity or expression has been implicated in various diseases, particularly cancer, where it supports cancer cell growth and resistance to energy stress-induced apoptosis by enhancing lipogenesis and upregulating oxidative phosphorylation (OXPHOS) (Fernandez et al, 2018; Mosaoa et al, 2021; Yang et al, 2021). However, the potential role of SLC25A1 in ferroptosis vulnerability remains unknown. In this study, we illustrate that SLC25A1, in coordination with ACLY, facilitates FSP1 acetylation, thus protecting FSP1 from proteasomal degradation. We also demonstrate that pharmacological blockade of SLC25A1 and ACLY increases cell susceptibility to ferroptosis and markedly enhances ferroptosis-induced tumor suppression. Therefore, our findings propose that the SLC25A1-ACLY pathway could be targeted to leverage ferroptosis for therapeutic intervention in cancer.

## Results

### Identification of SLC25A1 as a critical regulator of ferroptosis sensitivity

Several proteins within the SLC superfamily have been identified as regulators of ferroptosis, such as the cystine/glutamate transporter system (System $x_c^-$) comprised of SLC7A11 and SLC3A2, which plays a crucial role in the extracellular uptake of cystine in exchange of glutamate, thereby facilitating the synthesis of glutathione (GSH) and protecting cells against oxidative stress (Chen et al, 2021b; Koppula et al, 2021). To explore the involvement of additional SLC superfamily proteins in ferroptosis regulation, we employed a CRISPR/Cas9-mediated knockout screening targeting human SLC genes in A375 melanoma cells, using the GPX4 inhibitor RSL3 to induce ferroptosis (Fig. 1A). The screening validated the role of SLC7A11, which emerged as the most significantly enriched gene in the negative regulation of ferroptosis, with SLC3A2 also showing enrichment, thereby

confirming the robustness of our screening in identifying ferroptosis mediators. Notably, SLC25A1 was distinguished as the most enriched gene among those not previously associated with ferroptosis suppression (Fig. 1B). To evaluate the role of SLC25A1 as a ferroptosis inhibitor, we silenced its expression in A375 and A549 lung cancer cells and observed increased sensitivity to ferroptosis triggered by RSL3, erastin, and cystine deprivation (Fig. 1C–E; Appendix Fig. S1A–D). Further experiments extending the knockdown of SLC25A1 to SK-MEL-28, SK-MEL-30, HT29, RKO, Huh-7 and HepG2 cells confirmed this effect (Appendix Fig. S1E–G). In contrast, overexpressing SLC25A1 in SLC25A1-deficient A375 and A549 cells significantly attenuated RSL3- and erastin-triggered ferroptosis (Fig. 1F–H; Appendix Fig. S1H–J). We further observed that the ferroptosis inhibitor ferrostatin-1 (Fer-1) effectively mitigated the increase in RSL3-induced cell death and lipid ROS accumulation following SLC25A1 knockdown in A375 and A549 cells (Fig. 1I–L; Appendix Fig. S1K–N). Interestingly, the SLC25A1 knockdown alone induced cell death and increased lipid ROS levels (Appendix Fig. S1O,P), implying that SLC25A1 may directly induce ferroptosis in cancer cells. It has been reported that SLC25A1 promotes cancer cell survival and growth (Fernandez et al, 2018; Rochette et al, 2020; Yang et al, 2021). Consistent with previous studies, we confirmed that SLC25A1 knockdown significantly inhibited cell viability of A375 and A549 cells (Appendix Fig. S1Q). To determine the types of cell death induced by SLC25A1, we treated SLC25A1-depleted A375 and A549 cells with Fer-1 (ferroptosis inhibitor), Z-VAD-fmk (Z-V, apoptosis inhibitor), Necrostatin-1 (Nec-1, necrosis inhibitor), and Bafilomycin A1 (Baf-A1, autophagy inhibitor). Both Z-V and Fer-1 partially but significantly countered the cell death induced by SLC25A1 silencing, while Nec-1 and Baf-A1 showed no effect (Appendix Fig. S1R), suggesting that SLC25A1 knockdown directly induced cell apoptosis and ferroptosis. Further experiments using different cell death inhibitors on RSL3-treated parental and SLC25A1-depleted A375 and A549 cells revealed that only Fer-1 substantially prevented RSL3-induced cell death stimulated by SLC25A1 depletion, highlighting the significant impact of SLC25A1 on cancer cell sensitivity to ferroptosis (Fig. 1M;

Appendix Fig. S1S). Together, these results indicate that SLC25A1 is a critical regulator of ferroptosis.

Liver IRI is known to be associated with ferroptosis (Wu et al, 2021). To explore the role of SLC25A1 in modulating ferroptosis sensitivity in vivo, we generated liver-specific Slc25a1 knockdown (Slc25a1-KD) mice three weeks after tail vein injection using AAV8 virus and subsequently established a liver IRI model. Hematoxylin and eosin (H&E) staining demonstrated that while Slc25a1 knockdown alone did not induce liver injury, the combination of Slc25a1-KD and IRI significantly intensified inflammatory cell infiltration and hepatocyte death relative to IRI alone, and ferroptosis inhibitor Liproxstatin-1 (Lip-1) treatment protected against the liver injury (Fig. 1N). Immunohistochemistry (IHC) analysis further confirmed that Slc25a1 knockdown in conjunction with IRI markedly increased malondialdehyde (MDA) accumulation, a marker of oxidative stress, which was effectively inhibited by Lip-1 (Fig. 1N). In addition, our results showed that Slc25a1-KD + IRI significantly elevated the mRNA levels of ferroptosis markers, Ptgs2 and Chac1, in the liver, along with liver injury markers alanine aminotransferase (ALT) and aspartate amino-transferase (AST) in the serum, compared to IRI alone. Remarkably, these changes were completely reversed upon Lip-1 treatment (Appendix Fig. S2A,B). Together, these findings underscore the protective role of SLC25A1 against ferroptosis in vivo.

## Breakdown of citrate to acetyl-CoA by SLC25A1 and ACLY inhibits ferroptosis

Next, we delved into the mechanism by which SLC25A1 modulates cell sensitivity to ferroptosis. SLC25A1 facilitates the transport of citrate from mitochondria to the cytoplasm. Upon SLC25A1 depletion, citrate accumulation was observed in the mitochondria, with a concurrent reduction in the cytosol and nucleus (Fig. 2A; Appendix Fig. S3A). Our initial investigation focused on whether mitochondrial citrate metabolism influences ferroptosis suscept-ibility. Within mitochondria, citrate synthase (CS) catalyzes the formation of citrate from oxaloacetate (OAA) and acetyl-CoA, while aconitase 2 (ACO2) mediates the reversible conversion of citrate to isocitrate via cis-aconitate. We knocked down CS and ACO2 in A375 and A549 cells (Appendix Fig. S3B) and found that their depletion did not alter cell sensitivity to RSL3- and erastin-induced ferroptosis (Appendix Fig. S3C,D).

SLC25A1 is pivotal in maintaining the redox balance by facilitating NADPH production (Jiang et al, 2016; Zhang et al, 2023). SLC25A1 depletion was observed to elevate the NADP$^+$/NADPH ratio, leading to reduced glutathione (GSH) synthesis, potentially triggering ferroptosis (Appendix Fig. S3E,F). In the cytosol, ATP citrate lyase (ACLY) converts citrate into oxaloacetate and acetyl-CoA. SLC25A1 knockdown reduced acetyl-CoA levels in A375 and A549 cells (Fig. 2B). We found that ACLY knockdown heightened cell susceptibility to RSL3- and erastin-induced ferroptosis across various cancer cell lines, including A375, A549, RKO, and SK-MEL-30 (Fig. 2C–E; Appendix Fig. S3G–I). However, Fer-1 reversed the increase in RSL3-induced cell death attributable to ACLY knockdown (Fig. 2F). Similar to SLC25A1, ACLY depletion also led to lipid ROS accumulation and cell death, which Fer-1 and Z-V partially mitigated, implicating both ferroptosis and apoptosis (Appendix Fig. S3J–L). To confirm the critical role of cytosolic citrate conversion to acetyl-CoA in ferroptosis regulation

mediated by SLC25A1 and ACLY, we sequentially knocked down ACLY and SLC25A1. This did not exacerbate RSL3- and erastin-induced ferroptosis in A375 and A549 cells (Fig. 2G–I). Intrigu-ingly, ACLY knockdown upregulated SLC25A1 expression, hinting at a compensatory mechanism (Fig. 2G).

In addition to SLC25A1-ACLY axis, which regulates intracel-lular levels of acetyl-CoA, ACSS1/2 and pyruvate dehydrogenase complex (PDC) also can use acetate or pyruvate to generate acetyl-CoA (Guertin and Wellen, 2023). To determine whether ACSS1/2 and PDC are involved in the modulation of ferroptosis sensitivity by SLC25A1, we knocked down ACSS1/2 and E1 subunit of the PDC (PDHA1) in A375 cells, and found that their depletion did not affect sensitivity to RSL3-induced ferroptosis (Appendix Fig. S4A–F). In addition, in SLC25A1- and ACLY-depleted A375 cells, silencing ACSS1/2 and PDHA1 did not affect the enhanced ferroptosis sensitivity induced by SLC25A1 or ACLY depletion (Appendix Fig. S4G–R). These results indicate that SLC25A1/ACLY-mediated ferroptosis sensitivity are independent of the ACSS1/2- or PDC-mediated acetate or pyruvate metabolism pathway.

ACSS-dependent acetyl-CoA production is primarily utilized under metabolic stress conditions, such as hypoxia, lipid depletion or ACLY inhibition (Schug et al, 2015; Zhao et al, 2016). Consistently, sodium acetate supplementation alleviated the enhanced ferroptosis induced by SLC25A1 or ACLY deletion (Appendix Fig. S4S–W). Meanwhile, supplementation with dimethyl citrate rescued the increased ferroptosis caused by SLC25A1 knockdown (Appendix Fig. S4T–U). These findings collectively demonstrate that SLC25A1 and ACLY suppress ferroptosis through the breakdown of citrate to produce acetyl-CoA.

## FSP1 is acetylated at lysine 168, leading to increased protein stability

Subsequently, we investigated the impact of acetyl-CoA on cellular ferroptosis sensitivity. Given its role in de novo lipogenesis, we performed untargeted lipidomics analysis and observed a pro-nounced reduction in lipogenesis across various carbon chains following SLC25A1 knockdown (Appendix Fig. S5A). Previous research highlights that phosphatidylcholine (PC) and phosphati-dylethanolamine (PE) species, particularly those containing arachi-donic (AA, 20:4) or adrenic acid (AdA, 22:4) acyl chains, are essential polyunsaturated fatty acid phospholipids (PUFA-PLs) and preferred substrates for phospholipid peroxidation, significantly influencing ferroptosis sensitivity (Liang et al, 2023; Zheng and Conrad, 2020). Our analysis revealed a decrease in the vast majority of PE and PC species following SLC25A1 knockdown with no effect on total monounsaturated fatty acids (MUFA)/PUFA, PC-MUFA/PC-PUFA or PE-MUFA/PE-PUFA ratios (Appendix Fig. S5B,C), suggesting that SLC25A1's role in modulating ferroptosis sensitivity is not directly linked to PUFA-PL content.

FSP1 is a critical regulator against ferroptosis, and its myristoylation at the G2 residue is essential for maintaining its localization to the plasma membrane and enabling its anti-lipid peroxidation activity (Doll et al, 2019). Myristic acid (MA) is the only donor for protein myristoylation. To assess the impact of SLC25A1 and ACLY knockdown on MA availability, we quantified MA levels in cells using LC-MS/MS. Our analysis revealed that,

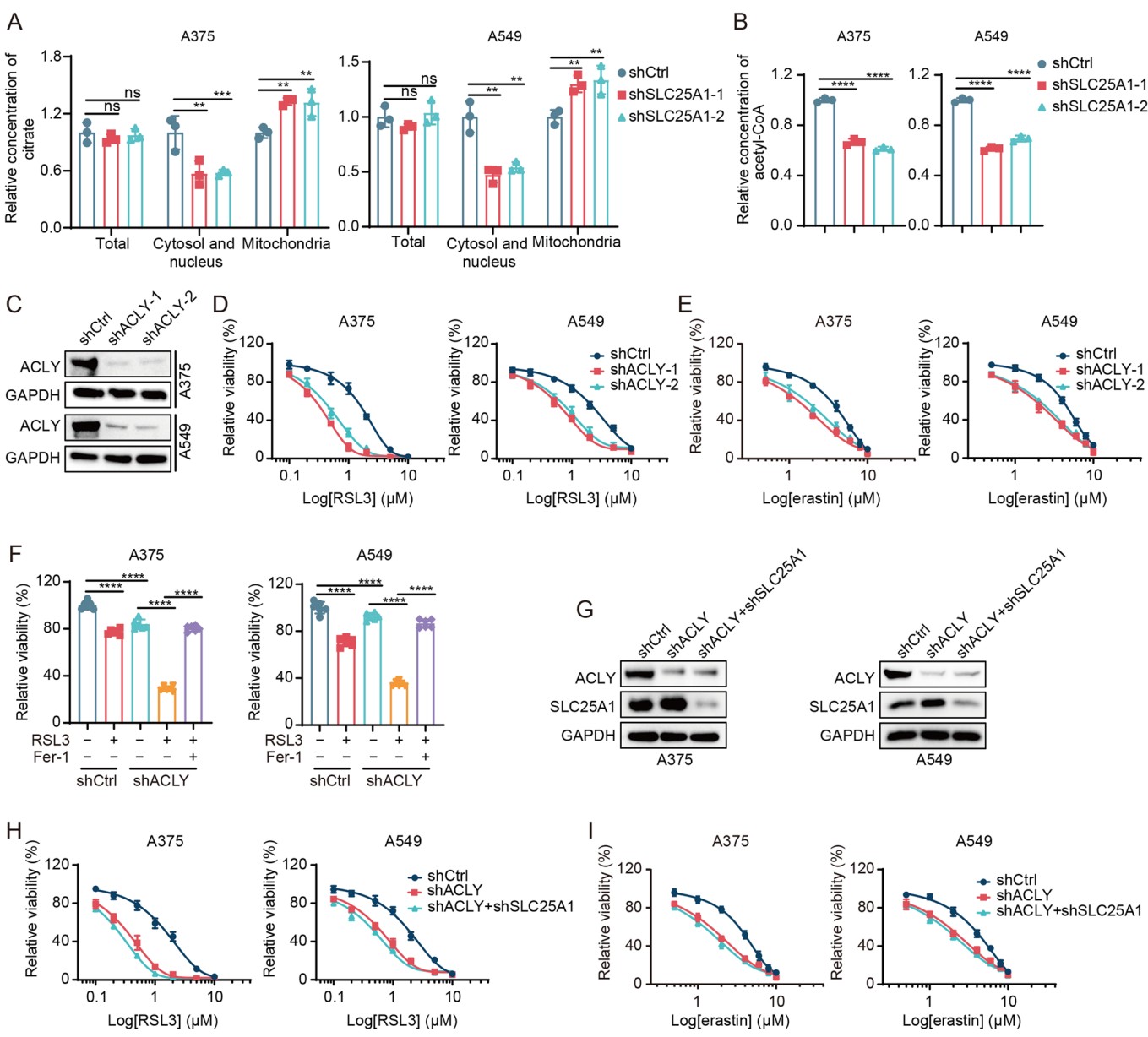

**Figure 2. Breakdown of citrate to acetyl-CoA by SLC25A1 and ACLY inhibits ferroptosis.**

(A) Measurement of citrate levels in the total cell, cytosol and nucleus, and mitochondria. n = 3, n represents biological independent experiments. Statistical analysis by two-way ANOVA tests; mean + SD, p values from left to right: ns: not significant, **p = 0.0092, **p = 0.0015, **p = 0.0012, **p = 0.0020 (A375 cells); ns: not significant; **p = 0.0045, **p = 0.0058, **p = 0.0066, **p = 0.0079 (A549 cells). (B) Measurement of cellular acetyl-CoA concentrations. n = 3, n represents biological independent experiments. Statistical analysis by two-tailed, unpaired Student's t-test; mean + SD, p values from left to right: ****p = 4.67E−05, ****p = 1.00E−05, ****p = 9.72E−06, ****p = 5.77E−05. (C) Immunoblot confirming the knockdown of ACLY in shACLY-A375/A549 cells. (D, E) Measurement of cell viability in shCtrl and shACLY-A375/A549 cells following treatment with incremental doses of RSL3 (D) or erastin (E) over 48 h. n = 3 for (D, E), n represents biological independent experiments. (F) Comparison of cell viability between shCtrl and shACLY-A375/A549 cells exposed to indicated treatments for 48 h. RSL3 at 0.5 μM, Fer-1 at 1 μM. n = 3, n represents biological independent experiments. Statistical analysis by two-way ANOVA tests; mean + SD, p values from left to right: ****p = 1.16E−07, ****p = 2.21E−05, ****p = 5.25E−11, ****p = 4.37E−12 (A375 cells); ****p = 3.75E−07, ****p = 4.97E−12, ****p = 4.40E−11 (A549 cells). (G) Immunoblot of ACLY and SLC25A1 protein levels in the indicated cells. Following ACLY knockdown, A375 and A549 cells were subjected to SLC25A1 knockdown (shACLY + shSLC25A1-A375/A549). (H, I) Cell viability assays for shCtrl, shACLY, and shACLY + shSLC25A1-A375/A549 cells treated with increasing RSL3 (H) or erastin (I) concentrations for 48 h. n = 3 for (H, I), n represents biological independent experiments. Source data are available online for this figure.

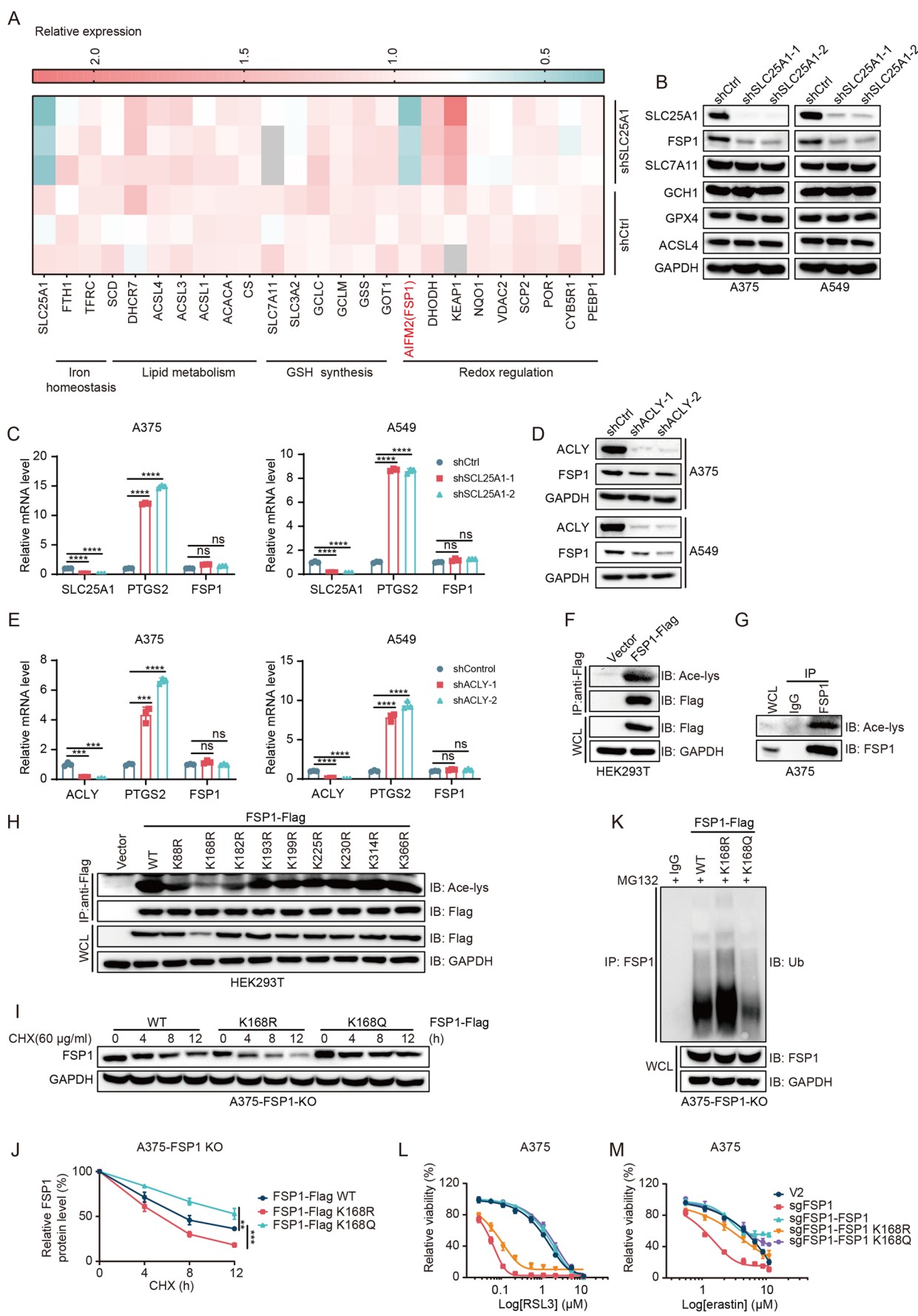

◀ **Figure 3. FSP1 is acetylated at lysine 168, leading to increased protein stability.**

(A) Heat map representing expression of proteins involved in ferroptosis in shCtrl and shSLC25A1-A375 cells from proteomics. (B) Immunoblot of the protein levels of SLC25A1, FSP1, SLC7A11, GCH1, GPX4, and ACSL4 in shCtrl versus shSLC25A1-silenced A375/A549 cells. (C) Quantitative RT-PCR (qRT-PCR) analysis of mRNA expressions of SLC25A1, PTGS2, and FSP1 in shCtrl and shSLC25A1-A375/A549 cells. $n = 3$, $n$ represents biological independent experiments. Statistical analysis by two-way ANOVA tests; mean + SD, $p$ values from left to right: ****$p = 2.53E-07$, ****$p = 2.27E-07$, ****$p = 1.99E-08$, ****$p = 8.65E-08$, ns: not significant (A375 cells); ****$p = 3.01E-05$, ****$p = 2.70E-05$, ****$p = 3.38E-08$, ****$p = 3.42E-07$, ns: not significant (A549 cells). (D) Immunoblot of ACLY and FSP1 protein levels in shCtrl and shACLY-silenced A375/A549 cells. (E) qRT-PCR analysis of mRNA expressions of ACLY, PTGS2, and FSP1 in shCtrl and shACLY-A375/A549 cells. $n = 3$, $n$ represents biological independent experiments. Statistical analysis by two-way ANOVA tests; mean + SD, $p$ values from left to right: ***$p = 0.00032$, ***$p = 0.00023$, ***$p = 0.00034$, ****$p = 1.08E-06$, ns: not significant (A375 cells); ****$p = 8.75E-06$, ****$p = 2.22E-06$, ****$p = 1.97E-05$, ****$p = 7.32E-06$, ns: not significant (A549 cells). (F) Immunoblot of acetylated FSP1-Flag in HEK293T cells, with empty vector as a negative control. (G) Immunoblot of acetylation of endogenous FSP1 in A375 cells. IgG was employed as a negative control. (H) Immunoblot of acetylation profiles of various FSP1-Flag mutants in HEK293T cells. (I, J) Immunoblot of FSP1 protein stability in A375-FSP1-KO cells re-expressing Flag-tagged FSP1 WT, K168R, or K168Q mutants, following cycloheximide (CHX, 60 μg/mL) treatment over specified durations (I). Quantification of FSP1 protein levels. $n = 3$, $n$ represents biological independent experiments. Statistical analysis by two-tailed, unpaired Student's t-test; mean + SD, $p$ values from left to right: **$p = 0.0094$, ***$p = 0.00023$ (J). (K) Immunoblot of FSP1 ubiquitination in A375-FSP1-KO cells stably expressing Flag-tagged FSP1 WT, K168R, or K168Q mutants, following 10 μM MG132 treatment for 12 h. IgG served as a negative control. (L, M) Measurement of cell viability in control sgRNA (V2), FSP1-knockout (sgFSP1), and FSP1-knockout A375 cells re-expressing FSP1 WT/K168R/K168Q exposed to increasing concentrations of RSL3 (L) or erastin (M) for 48 h. $n = 3$ for (L, M), $n$ represents biological independent experiments. Source data are available online for this figure.

although silencing SLC25A1 or ACLY significantly reduced palmitic acid (PA) levels, MA levels remained largely unchanged (Appendix Fig. S5D). This discrepancy may be attributed to the relatively low cellular abundance of MA, suggesting that its homeostasis could be tightly regulated. Therefore, the knockdown of SLC25A1 or ACLY does not appear to affect the myristoylation of FSP1.

Prior studies have indicated that inhibiting SLC25A1 and ACLY can diminish histone acetylation, affecting gene transcription (Li et al, 2022). We therefore performed RNA-seq analysis to determine whether SLC25A1 regulates ferroptosis at the transcriptional levels. KEGG pathway analysis revealed that SLC25A1 knockdown significantly upregulated several inflammatory and immune pathways, including TNF, IL-17, NK cell-mediated cytotoxicity, which inhibit cancer development. On the other hand, silencing SLC25A1 resulted in the downregulation of various lipid synthesis pathways, including those for thyroid hormones, cholesterol, and glycosphingolipids, alongside pathways involved in glucose and amino acid metabolism (Appendix Fig. S5E). However, it did not show enrichment for pathways directly related to ferroptosis. A targeted examination of ferroptosis-related genes revealed no significant changes in the transcription levels of critical genes such as GPX4, FSP1, SLC7A11, and ACSL4, although it upregulated PTGS2 and CHAC1 transcription, the marker of ferroptosis (Appendix Fig. S5F). These results suggested that knockdown of SLC25A1 does not induce ferroptosis by modulating the expression of specific ferroptosis-related genes.

Acetyl-CoA is the sole donor for protein acetylation, significantly influencing protein functions, including their activity, stability, and localization (Narita et al, 2019). Therefore, we performed proteomics and KEGG analysis, and observed that SLC25A1 knockdown affected several pathways, including pyruvate metabolism, fructose and mannose metabolism and TCA cycle, which was consistent with the RNA-seq results. Furthermore, knocking down SLC25A1 inhibited DNA damage repair, such as mismatch repair and nucleotide excision repair (Appendix Fig. S5G). Targeted analysis of ferroptosis-related proteins showed that knockdown of SLC25A1 was associated with a significant reduction in the protein level of FSP1, a key protein in ferroptosis, while the levels of other proteins such as ACSL4, SLC7A11 and DHODH were unchanged (Fig. 3A). We further determined that

silencing of SLC25A1 in A375 and A549 cells resulted in a significant reduction in FSP1 protein levels, whereas the protein levels of SLC7A11, GCH1, GPX4, and ACSL4, other important regulators of ferroptosis remained unchanged (Fig. 3B). Meanwhile, SLC25A1 knockdown did not alter FSP1 mRNA levels but significantly increased PTGS2 mRNA levels, which corresponded to the RNA-seq results (Fig. 3C). Similarly, ACLY knockdown in A375 and A549 cells resulted in reduced FSP1 protein levels and elevated PTGS2 mRNA levels, without affecting FSP1 mRNA levels (Fig. 3D,E). Overexpression of SLC25A1 not only enhanced FSP1 protein levels but also counteracted the reduction in FSP1 protein levels caused by SLC25A1 knockdown (Appendix Fig. S6A,B). These findings suggest that SLC25A1 and ACLY modulate FSP1 expression directly at the protein levels.

We further investigated whether FSP1 undergoes acetylation. We detected FSP1 acetylation upon overexpressing FSP1 in HEK293T cells (Fig. 3F). This finding was corroborated in A375 and A549 cells, confirming endogenous FSP1 acetylation (Fig. 3G; Appendix Fig. S6C). Given that lysine residues serve as sites for both acetylation and ubiquitination-often sharing overlapping functions, we employed GPS-PAIL 2.0 for site predictions and mutated each of the nine lysine residues of FSP1 to arginine. We identified K168 as the primary acetylation site on FSP1 (Fig. 3H), which was also previously detected in HeLa cells (Hansen et al, 2019). To elucidate the impact of FSP1 acetylation at K168 on protein stability, we introduced wild-type (WT) FSP1, an acetylation-resistant K168R mutant, and an acetylation-mimetic K168Q mutant into FSP1-knockout (KO) A375 and A549 cells. Cycloheximide (CHX) chase assays revealed that the K168R mutant exhibited a reduced protein half-life, whereas the K168Q mutant showed an extended half-life compared to WT FSP1 (Fig. 3I,J; Appendix Fig. S6D). Ubiquitination assays further demonstrated that the K168Q mutant displayed significantly reduced ubiquitination levels, in contrast to the heightened ubiquitination observed for the K168R mutant (Fig. 3K; Appendix Fig. S6E). Functionally, both WT FSP1 and the K168Q mutant mitigated the increased ferroptosis sensitivity seen in FSP1-KO cells, whereas the K168R mutant's protective effect was comparatively weaker (Fig. 3L,M). To ensure uniform transcriptional activity across variants, we verified that mRNA levels remained consistent (Appendix Fig. S6F). Moreover, we found that K168Q mutation did not influence FSP1's

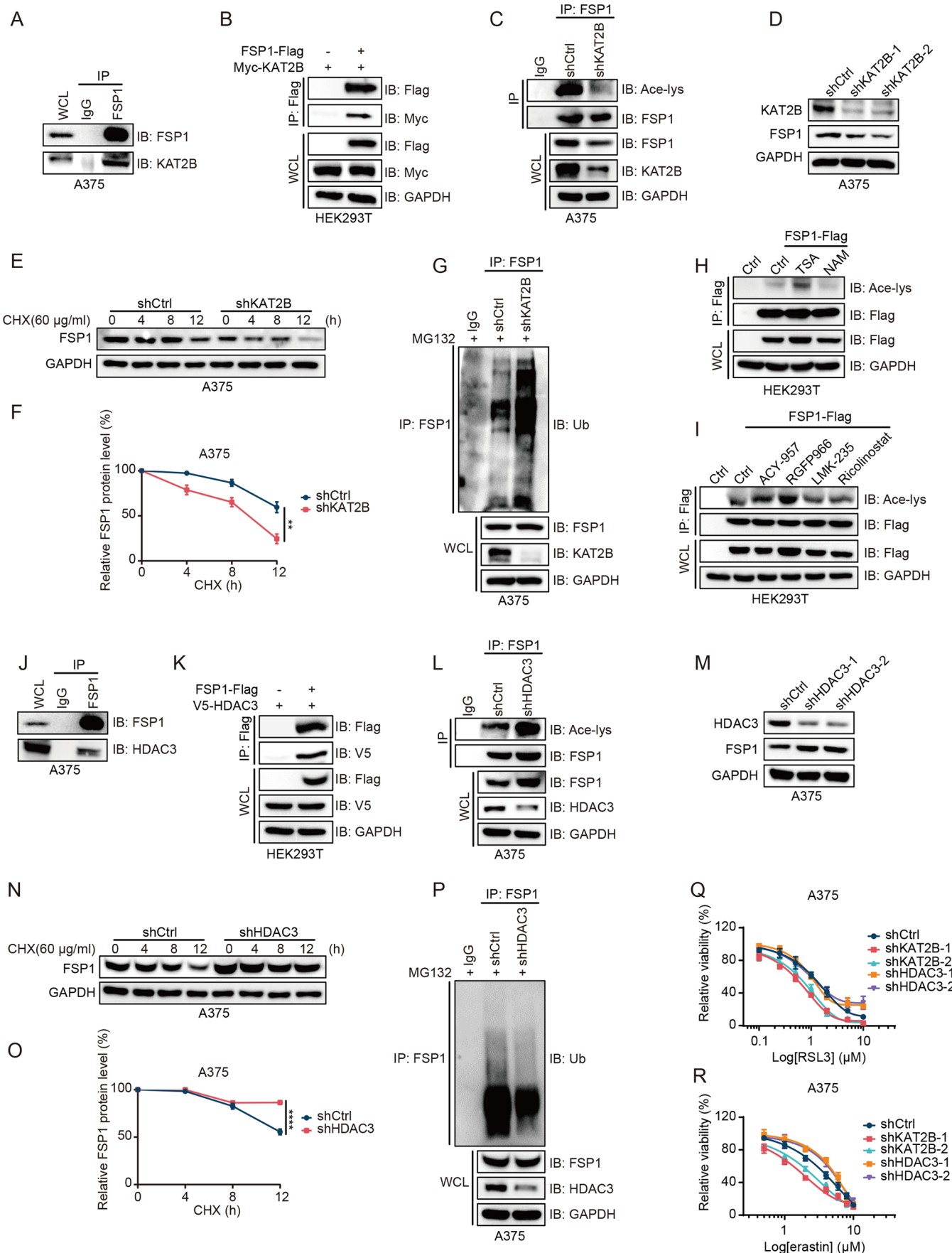

**Figure 4.   FSP1 acetylation and deacetylation is predominantly mediated by KAT2B/HDAC3.**

(A) Immunoblot detecting endogenous interaction between FSP1 and KAT2B in A375 cells, with IgG as a negative control for IP. (B) Immunoblot of exogenous interaction between FSP1-Flag and Myc-tagged KAT2B in HEK293T cells, with Flag empty vector as a negative control. (C) Immunoblot of FSP1 acetylation following KAT2B knockdown in A375 cells with IgG as a negative control. (D) Immunoblot of FSP1 protein levels in shCtrl versus shKAT2B-A375 cells. (E, F) Time-dependent FSP1 protein stability assessed via immunoblot in shCtrl and shKAT2B-A375 cells treated with CHX (60 μg/ml) for indicated durations and quantification of FSP1 protein levels (E). Quantification of FSP1 protein levels. $n = 3$, $n$ represents biological independent experiments. Statistical analysis by two-tailed, unpaired Student's t-test; mean + SD, **$p = 0.0016$ (F). (G) Immunoblot of FSP1 ubiquitination in shCtrl and shKAT2B-A375 cells following treatment with MG132 (10 μM) for 12 h, with IgG as a negative control. (H) Immunoblot of acetylation of exogenous FSP1 upon treatment with HDAC inhibitors: trichostatin A (TSA, 1 μM, 12 h) or nicotinamide (NAM, 10 mM, 6 h) in HEK293T cells transfected with FSP1-Flag, with Flag empty vector as a negative control. (I) Immunoblot of the acetylation of exogenous FSP1 in HEK293T cells with indicated treatment: ACY957 (10 μM, 6 h), RGFP966 (1 μM, 24 h), LMK-235 (1 μM, 24 h), and Ricolinostat (40 μM, 24 h), with Flag empty vector as a negative control. (J) Immunoblot of endogenous interaction between FSP1 and HDAC3 in A375 cells, with IgG as a negative control. (K) Immunoblot of exogenous interaction between FSP1-Flag and V5-tagged HDAC3 in HEK293T cells, with Flag empty vector as a negative control. (L) Immunoblot of FSP1 acetylation following HDAC3 knockdown in A375 cells with IgG as a negative control. (M) Immunoblot of FSP1 protein levels in shCtrl versus shHDAC3-A375 cells. (N, O) Immunoblot of FSP1 protein stability in shCtrl and shHDAC3-A375 cells treated with CHX (60 μg/ml) for indicated times and quantification of FSP1 protein levels (N). $n = 3$, $n$ represents biological independent experiments. Statistical analysis by two-tailed, unpaired Student's t-test; mean + SD, ****$p = 8.39E−05$ (O). (P) Immunoblot of FSP1 ubiquitination in shCtrl and shHDAC3-A375 cells after MG132 treatment (10 μM, 12 h), with IgG as a negative control. (Q, R) Cell viability assays in shCtrl, shKAT2B, and shHDAC3-A375 cells treated with increasing concentrations of RSL3 (O) or erastin (P) for 48 h. $n = 3$ for (O–R), $n$ represents biological independent experiments. Source data are available online for this figure.

enzymatic activity (Appendix Fig. S6G). Immunofluorescence studies showed that neither mutation significantly altered FSP1's cellular localization (Appendix Fig. S6H), indicating that the anti-ferroptotic function of FSP1 is primarily dictated by its stability rather than its activity or localization. Collectively, these results highlight K168 as the primary acetylation site on FSP1, crucial for modulating its protein stability and role in counteracting ferroptosis.

## KAT2B/HDAC3 predominantly mediates FSP1 acetylation/deacetylation

Acetylation, a dynamic and reversible post-translational modification, is modulated by acetyltransferases and deacetylases. GPS-PAIL 2.0 predicted that KAT2B induces acetylation at the K168 site of FSP1. To test this hypothesis, we examined the interaction between endogenous FSP1 and KAT2B in A375 and A549 cells and confirmed their association (Fig. 4A; Appendix Fig. S7A). This interaction was also observed with ectopically expressed FSP1 and KAT2B in HEK293T cells (Fig. 4B). Subsequent knockdown of KAT2B in A375 and A549 cells significantly diminished FSP1 acetylation and reduced FSP1 protein levels (Fig. 4C,D; Appendix Fig. S7B,C). Notably, KAT2B did not enhance the acetylation of the FSP1 K168Q mutant, indicating specificity to the acetylation site (Appendix Fig. S7D). Furthermore, KAT2B knockdown led to accelerated degradation and increased ubiquitination of FSP1 (Fig. 4E–G; Appendix Fig. S7E,F). These findings suggest that KAT2B acts as the acetyltransferase for the acetylation of FSP1 at the K168 site. To identify the deacetylases involved in FSP1 deacetylation, we treated HEK293T cells expressing exogenous FSP1 with trichostatin A (TSA), an inhibitor of the HDAC family, or nicotinamide (NAM), an inhibitor of the SIRT family deacetylases. Increased FSP1 acetylation was observed in cells treated with TSA but not with NAM (Fig. 4H). Further experiments with specific inhibitors-ACY957 for HDAC1/2, RGFP966 for HDAC3, LMK-235 for HDAC4/5, and Ricolinostat for HDAC6-revealed that only RGFP966 treatment led to enhanced FSP1 acetylation (Fig. 4I), indicating HDAC3 as the primary deacetylase for FSP1. Both endogenous and exogenous immunoprecipitation (IP) assays confirmed the interaction between FSP1 and HDAC3

(Fig. 4J,K; Appendix Fig. S7G). Knockdown of HDAC3 not only increased FSP1's acetylation but also its protein levels (Fig. 4L,M; Appendix Fig. S7H,I). In addition, HDAC3 could not further decrease the acetylation of FSP1 mutants at K168R (Appendix Fig. 7J), suggesting specificity towards the acetylated lysine. With HDAC3 silenced, we observed an increase in FSP1 protein stability and a decrease in its ubiquitination (Fig. 4N–P; Appendix Fig. S7K,L), underscoring HDAC3's pivotal role in mediating FSP1 K168 deacetylation. In general, the ubiquitin chains formed by linking K48, K29 and K11 are used as tags for substrate proteins to be targeted to the proteasome for degradation (Gao et al, 2021; Kaiho-Soma et al, 2021; Tracz and Bialek, 2021). Using a series of HA tagged ubiquitin mutants (K11R, K29R and K48R), we found only the mutation of K29 disrupted FSP1 ubiquitination (Appendix Fig. S7M). Consistently, K29-only ubiquitin supported FSP1 ubiquitination, which was abolished by KAT2B, whereas HDAC3 further promoted K29-linked ubiquitination of FSP1 (Appendix Fig. S7N,O). These results confirmed that the ubiquitination interplay with acetylation regulates the protein stability of FSP1, in which KAT2B and HADC3 play a critical role. Correlating with these findings, cells with KAT2B knockdown exhibited increased sensitivity to ferroptosis, whereas HDAC3 knockdown conferred slight resistance, likely due to the presence of sufficient FSP1 protein to counteract RSL3- and erastin-induced ferroptosis (Fig. 4Q,R).

## SLC25A1 and ACLY regulate FSP1 acetylation and stability

To assess whether SLC25A1 and ACLY influence FSP1 acetylation, we silenced both genes and observed a significant reduction in FSP1 acetylation and protein levels (Fig. 5A,B; Appendix Fig. S8A,B). Conversely, supplementation with sodium acetate partially restored the reduced FSP1 acetylation levels (Appendix Fig. S8C,D). This knockdown also markedly increased FSP1 ubiquitination and expedited the degradation of FSP1 protein in A375 and A549 cells (Fig. 5C–F; Appendix Fig. S8E–H). In addition, we performed subcellular fractionation to isolate plasma membrane and cytosolic fractions, which demonstrated that knockdown of SLC25A1 or ACLY did not affect the plasma membrane localization of FSP1

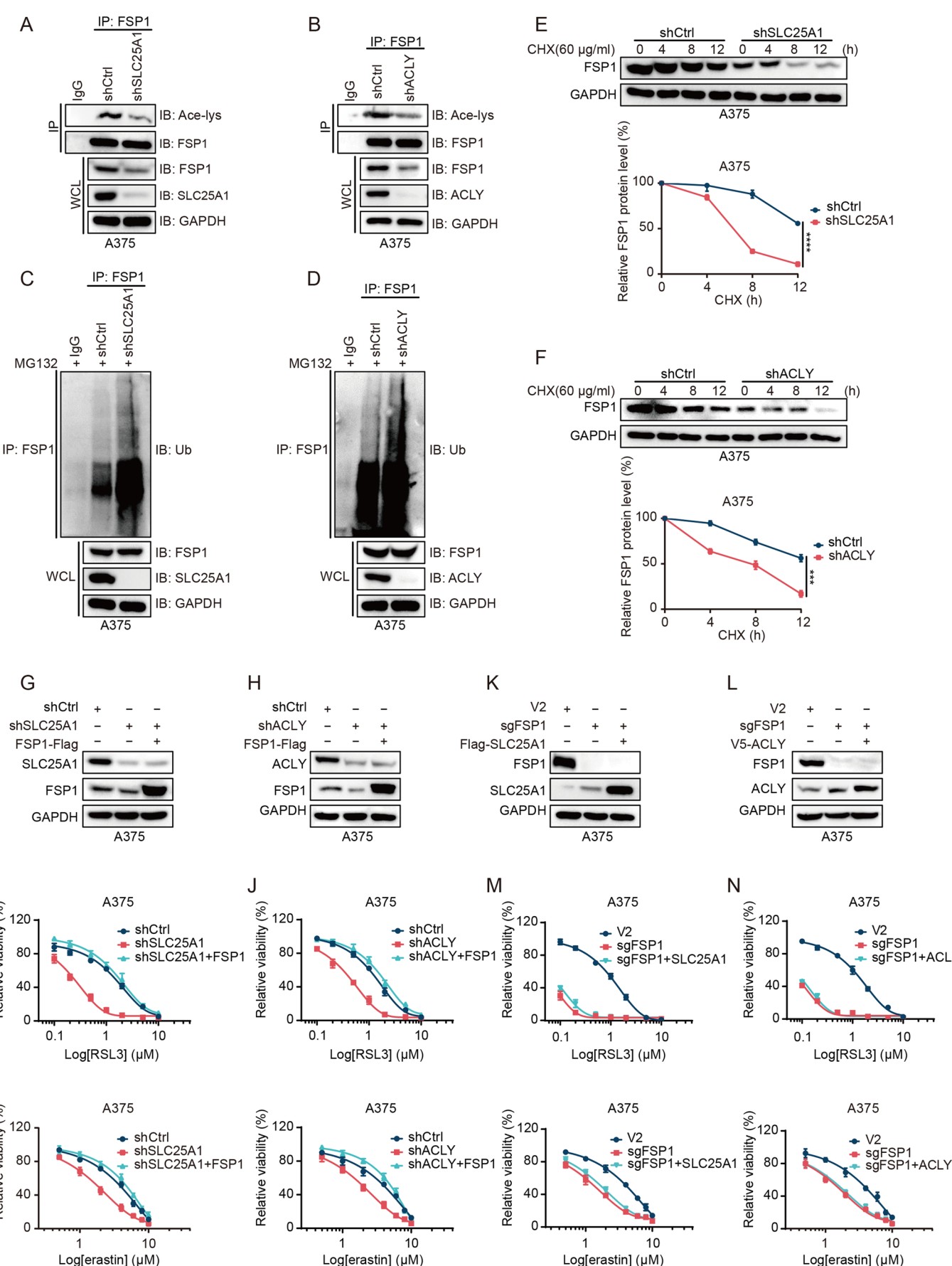

**Figure 5. SLC25A1 and ACLY regulate FSP1 acetylation and stability.**

(**A, B**) Immunoblot of FSP1 acetylation levels in A375 cells stably transfected with lentiviral shRNA targeting SLC25A1 (**A**), ACLY (**B**), or a control shRNA. IgG served as a negative control. (**C, D**) Immunoblotting of ubiquitination of FSP1 in A375 cells stably expressing shRNA against SLC25A1 (**C**), ACLY (**D**), or control shRNA, following 10 μM MG132 treatment for 12 h. IgG served as a negative control. (**E, F**) Immunoblots of FSP1 protein levels in A375 cells with stable shRNA-mediated knockdown of SLC25A1 (**E**), ACLY (**F**), or control shRNA, treated with CHX (60 μg/ml) for the indicated durations and quantification of FSP1 protein levels. $n = 3$ for (**E, F**), $n$ represents biological independent experiments. Statistical analysis by two-tailed, unpaired Student's t-test; mean + SD, ****$p = 1.10E-05$ (**E**), ***$p = 0.00021$ (**F**). (**G, H**) Immunoblots of SLC25A1, ACLY, and FSP1 in A375 cells following knockdown of SLC25A1 (**G**) or ACLY (**H**) and overexpression of FSP1. (**I, J**) Cell viability assays of A375 cells expressing control shRNA (shCtrl), shRNA against SLC25A1 (shSLC25A1), shSLC25A1 with FSP1 overexpression (shSLC25A1 + FSP1), shRNA against ACLY (shACLY), and shACLY with FSP1 overexpression (shACLY + FSP1) treated with increasing concentrations of RSL3 (**I**) or erastin (**J**) for 48 h. $n = 3$ for (**I, J**), $n$ represents biological independent experiments. (**K, L**) Immunoblot of FSP1, SLC25A1, and ACLY levels in FSP1-knockout A375 cells overexpressing SLC25A1 (**K**) or ACLY (**L**). (**M, N**) Cell viability assays of A375 cells transfected with V2, sgRNA targeting FSP1 (sgFSP1), sgFSP1 with SLC25A1 overexpression (sgFSP1 + SLC25A1), and sgFSP1 with ACLY overexpression (sgFSP1 + ACLY) exposed to increasing concentrations of RSL3 (**M**) or erastin (**N**) for 48 h. $n = 3$ for (**M, N**), $n$ represents biological independent experiments. Source data are available online for this figure.

(Appendix Fig. S8I,J). Consistently, the immunofluorescence analyses indicated that while knockdown of SLC25A1 or ACLY reduced FSP1 expression, it did not alter the subcellular localization of FSP1 (Appendix Fig. S8K). Together, these results indicate that SLC25A1 or ACLY affects FSP1 protein stability but not its localization.

To further verify that SLC25A1 and ACLY contribute to ferroptosis resistance by stabilizing FSP1, we knocked down SLC25A1 or ACLY and subsequently overexpressed FSP1. The overexpression of FSP1 notably mitigated the sensitivity of A375 and A549 cells to RSL3- and erastin-induced ferroptosis (Fig. 5G–J; Appendix Fig. S9A–D). Meanwhile, we introduced the acetylation-mimetic K168Q mutant and the non-acetylatable K168R mutant into FSP1-knockout A549 cells with or without depletion of SLC25A1 or ACLY. Our findings indicated that the FSP1 K168Q mutant effectively rescued the increased sensitivity to ferroptosis observed in FSP1-KO cells, while the K168R mutant exhibited a notably weaker protective effect, due to its reduced protein stability (Appendix Fig. S9E,F). Importantly, regardless of the FSP1 mutants, the knockdown of SLC25A1 or ACLY did not further enhance cellular sensitivity to RSL3-induced ferroptosis (Appendix Fig. S9E,F). Conversely, overexpressing SLC25A1 or ACLY in FSP1-knockout cells did not effectively diminish the cells' ferroptosis sensitivity (Fig. 5K–N; Appendix Fig. S9G–J). Intriguingly, FSP1 knockout slightly elevated SLC25A1 and ACLY protein levels, with qPCR results indicating this might be due to FSP1 knockout enhancing SLC25A1 and ACLY transcription, suggesting a negative feedback mechanism (Fig. 5K,L; Appendix Fig. S9K,L). Furthermore, although silencing of SLC25A1 was associated with a reduction in most lipid species, knockdown of SLC25A1 or ACLY moderately upregulated CoQ10 content, suggesting that this might be due to a reduction in FSP1 protein levels (Appendix Fig. S9M). These findings elucidate that SLC25A1 and ACLY suppress ferroptosis primarily by preserving FSP1 acetylation, thereby promoting FSP1 protein stability.

## Targeting SLC25A1 and ACLY increases ferroptosis sensitivity in vivo

We next examined whether pharmacological targeting of SLC25A1 and ACLY could be exploited for the modulation of ferroptosis. In A375 and A549 cells, treatment with the SLC25A1 inhibitor BTA or the ACLY inhibitor SB204990 significantly enhanced sensitivity to RSL3- and erastin-induced ferroptosis (Fig. 6A,B). The ferroptosis

inhibitor Fer-1 effectively reversed the increased cell death caused by these inhibitors (Fig. 6C,D). Mirroring the results from SLC25A1 and ACLY knockdown, both BTA and SB204990 dose-dependently diminished FSP1 protein levels (Fig. 6E), while supplementation with dimethyl citrate and sodium acetate restored FSP1 protein abundance (Fig. 6F).

To evaluate the impact of BTA and SB204990 on ferroptosis in vivo, we conducted liver IRI experiments. H&E staining revealed that both BTA and SB204990 intensified liver damage, showing widespread hepatocellular rupture and inflammation. IHC further showed that treatment with BTA and SB204990 increased MDA positivity while reducing FSP1 positivity in liver tissues, effects that were mitigated by Lip-1 (Fig. 6G; Appendix Fig. S10A). In addition, BTA specifically reduced Slc25a1 mRNA levels, and both inhibitors upregulated the transcription of ferroptosis markers Ptgs2 and Chac1 alongside a significant rise in serum ALT and AST levels, which could be attenuated by Lip-1 (Fig. 6H,I; Appendix Fig. S10B,C).

Further linking the inhibition of the SLC25A1/ACLY-FSP1 pathway to cancer, we investigated the effect of BTA or SB204990 on tumor sensitivity to ferroptosis in the A375 xenograft model. Specifically, combining BTA or SB204990 with the ferroptosis inducer IKE markedly inhibited tumor growth and weight, an effect that was significantly reduced by Lip-1, indicating the enhanced anti-tumor effect was mediated through ferroptosis (Fig. 6J–L). In addition, the combination treatment decreased the presence of Ki67- and FSP1-positive cells in the tumor more than IKE alone, with increased MDA accumulation, effects reversible by Lip-1 (Appendix Fig. S10D). These findings demonstrate that pharmacological inhibition of SLC25A1 or ACLY sensitizes tumors to ferroptosis, underscoring a potential therapeutic strategy targeting this cell death pathway in cancer treatment.

## The high expression SLC25A1-ACLY axis predicts the poor prognosis of cancer patients

We conducted further analysis on the relationship between SLC25A1 and FSP1 protein levels using the DepMap database. Proteomics analysis from the Cancer Cell Line Encyclopedia revealed a positive correlation between SLC25A1 and FSP1 expression (Fig. 7A). To further validate the clinical relevance of our findings, we analyzed a tissue microarray (TMA) comprising 111 tumor samples from melanoma patients to assess the expression of SLC25A1, ACLY, and FSP1 using multiplex

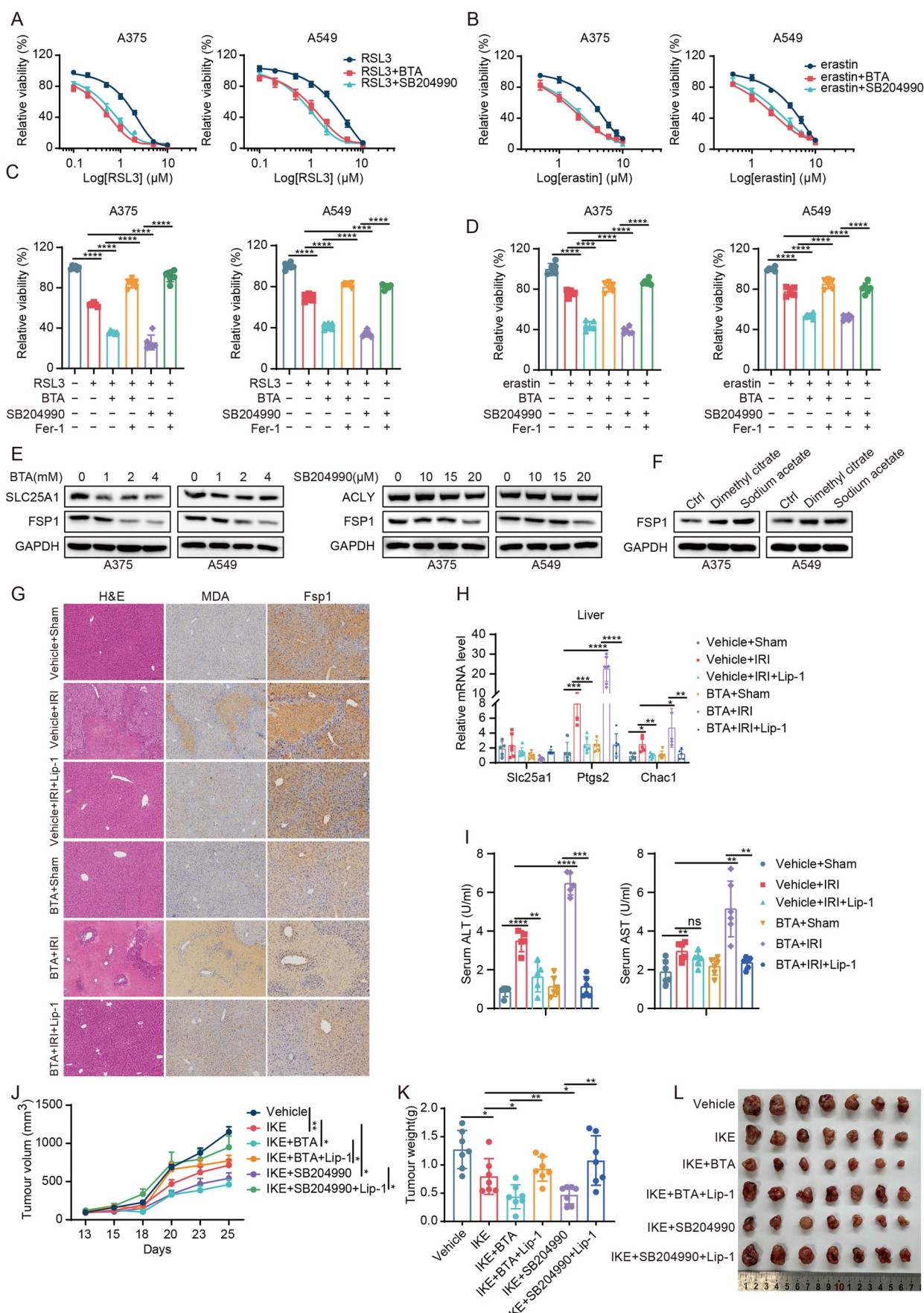

**Figure 6. Targeting SLC25A1 and ACLY increases ferroptosis sensitivity in vivo.**

(A, B) Cell viability assays of A375 and A549 cells treated with increasing concentrations of RSL3 (A) or erastin (B), with and without the addition of BTA (5 mM) or SB204990 (20 µM). $n = 3$ for (A, B), $n$ represents biological independent experiments. (C, D) Cell viability assays of A375 and A549 cells subjected to indicated treatments for 48 h: RSL3 (0.5 µM), erastin (1 µM), BTA (5 mM), SB204990 (20 µM), and Ferrostatin-1 (Fer-1, 1 µM). $n = 3$ for (C, D), $n$ represents biological independent experiments. Statistical analysis by two-way ANOVA tests; mean + SD, $p$ values from left to right: ****$p = 3.70E-12$, ****$p = 1.57E-11$, ****$p = 1.25E-10$, ****$p = 2.30E-07$, ****$p = 7.20E-09$ (C: A375 cells); ****$p = 1.60E-08$, ****$p = 3.67E-08$, ****$p = 8.46E-11$, ****$p = 2.92E-09$, ****$p = 2.80E-11$ (C: A549 cells); ****$p = 3.06E-06$, ****$p = 3.49E-08$, ****$p = 3.64E-08$, ****$p = 1.64E-09$, ****$p = 7.60E-11$ (D: A375 cells); ***$p = 3.19E-07$, ****$p = 2.33E-07$, ****$p = 3.94E-08$, ****$p = 1.51E-07$, ****$p = 2.25E-07$ (D: A549 cells). (E) Immunoblot of FSP1 protein levels in A375 and A549 cells treated with BTA or SB204990. (F) Immunoblot of FSP1 protein levels in A375 and A549 cells treated with dimethyl citrate or sodium acetate. (G) Mice were injected with vehicle, BTA (50 mg/kg) for three consecutive days. Lip-1 (10 mg/kg) was injected half an hour before ischemia reperfusion (IRI) or sham-treatment. Representative images showing H&E staining, MDA and Fsp1 immunohistochemical staining of livers from mice under the indicated treatment conditions. Scale bars: right, 200 nm. The experiment was repeated three times. (H) qRT-PCR analysis of mRNA expressions of Slc25a1, Ptgs2, and Chac1 from mice liver under the indicated treatment conditions. Treatments include Vehicle + Sham, Vehicle + IRI, Vehicle + IRI + Lip-1, BTA + Sham, BTA + IRI and BTA + IRI + Lip-1. $n = 5, 6, 7, 6, 7, 7$ (Slc25a1); 5, 5, 6, 5, 7, 6 (Ptgs2); 5, 5, 7, 6, 5, 5 (Chac1), $n$ represents biological independent experiments. Statistical analysis by two-way ANOVA tests; mean + SD, $p$ values from left to right: ***$p = 0.00052$, ***$p = 0.00094$, ****$p = 2.69E-05$, ****$p = 1.08E-05$ (Ptgs2); *$p = 0.011$, **$p = 0.0014$, *$p = 0.016$, **$p = 0.006$ (Chac1). (I) Measurement of ALT and AST levels in the serum from mice under the indicated treatment conditions. Treatments include Vehicle + Sham, Vehicle + IRI, Vehicle + IRI + Lip-1, BTA + Sham, BTA + IRI and BTA + IRI + Lip-1. $n = 5, 5, 6, 5, 6, 6$ (ALT); 6, 6, 6, 6, 6, 6 (AST), $n$ represents biological independent experiments. Statistical analysis by two-way ANOVA tests; mean + SD, $p$ values from left to right: ****$p = 1.09E-05$, ***$p = 0.0022$, ****$p = 3.19E-05$, ****$p = 6.36E-08$ (ALT), **$p = 0.0065$, ns: not significant, **$p = 0.0052$, *$p = 0.00094$ (AST). (J) Evaluation of tumor growth in xenograft mouse models using A375 cells. Upon reaching approximately 100 mm³, tumors were subjected to random division into six groups ($n = 7$ per group) and treated intraperitoneally (i.p.) every two days over 13 days with the following: (1) Vehicle, (2) IKE (50 mg/kg), (3) BTA (50 mg/kg) + IKE (50 mg/kg), (4) BTA (50 mg/kg) + IKE (50 mg/kg) + Lip-1 (10 mg/kg), (5) SB204990 (50 mg/kg) + IKE (50 mg/kg), (6) SB204990 (50 mg/kg) + IKE (50 mg/kg) + Lip-1 (10 mg/kg). $n = 7$, $n$ represents biological independent experiments. Statistical analysis by two-tailed, unpaired Student's t-test; mean + SD, $p$ values from left to right: **$p = 0.00098$, *$p = 0.048$, *$p = 0.039$, *$p = 0.042$, *$p = 0.0171$. (K) Tumor weights were measured at the end of the study on day 25. $n = 7$, $n$ represents biological independent experiments. Statistical analysis by two-tailed, unpaired Student's t-test; mean + SD, $p$ values from left to right: *$p = 0.0183$, *$p = 0.028$, **$p = 0.001$, *$p = 0.033$, **$p = 0.005$. (L) Representative images of tumors from each treatment group taken 25 days after inoculation. Source data are available online for this figure.

immunofluorescence (mIF) staining. Notably, increased expression of SLC25A1, ACLY was accompanied with elevated levels of FSP1 in the tumors (Fig. 7B). Further analysis confirmed a positive correlation between the expression of SLC25A1, ACLY and FSP1 (Fig. 7C,D). High expression of SLC25A1 or ACLY has been reported in multiple cancers including colorectal (Yang et al, 2021), breast (Chen et al, 2020), and lung cancer (Lin et al, 2013). Moreover, Kaplan–Meier survival analysis indicated that patients with high SLC25A1 and ACLY expression in skin cutaneous melanoma (SKCM) and lung adenocarcinoma (LUAD) had shorter survival than those with low SLC25A1 and ACLY expression (Fig. 7E,F). Taken together, these analyses indicate the clinical significance of SLC25A1-ACLY axis in prognosis of cancer patients.

## Discussion

SLC25A1 facilitates the transport of citrate from mitochondria to the cytoplasm, where it is converted by ACLY into acetyl-CoA, feeding into lipid synthesis and protein acetylation (Peng et al, 2020). The overexpression of SLC25A1, driven by mutations in tumor suppressor genes like *p53* and *PTEN*, is associated with poor prognosis in various cancers and enhances cancer cell resistance to energy stress (Albanese and Avantaggiati, 2015; Peng et al, 2020). Similarly to SLC25A1, ACLY is significantly elevated in a variety of cancers, including melanoma (Guo et al, 2020), hepatocellular carcinoma (Gu et al, 2021), lung cancer (Lin et al, 2013), prostate cancer (Li et al, 2019), and breast cancer (Adorno-Cruz et al, 2021). Phosphorylated ACLY expression was significantly associated with stage, differentiation and prognosis in patients with lung adeno-carcinoma (Migita et al, 2008), and high ACLY expression was associated with drug resistance, recurrence and poor prognosis in breast cancer (Chen et al, 2020). SLC25A1 and ACLY support rapid cell proliferation and epithelial-mesenchymal transition and

maintain cancer stem cell stemness by upregulating fatty acid synthesis and histone acetylation (Carrer et al, 2019; Hanai et al, 2013; Lee et al, 2018; Mosaoa et al, 2021; Wen et al, 2019). Inhibition of SLC25A1 or ACLY leads to impaired cancer cell proliferation, differentiation, and sensitization to immune therapy, highlighting their role as potential effective targets for anti-cancer strategy (Huang et al, 2022; Wartewig et al, 2023; Xiang et al, 2023; Zhang et al, 2023). Here, we elucidate that SLC25A1 and ACLY are pivotal in regulating sensitivity to ferroptosis. Intriguingly, their influence on ferroptosis does not directly stem from modulating lipid metabolism. Depleting SLC25A1 reduces the levels of most lipid species with minimal impact on PUFA-PE species, the critical substrates for phospholipid peroxidation and ferroptosis induction. Furthermore, SLC25A1 depletion does not alter the transcription of proteins related to ferroptosis, indicating that their regulation of ferroptosis sensitivity is not mediated through histone acetylation. Instead, SLC25A1 and ACLY elevate acetyl-CoA levels, enhancing the acetylation of FSP1, a crucial anti-ferroptosis mechanism alongside GPX4. Notably, the knockdown of SLC25A1 or ACLY alone can partially induce ferroptosis even without ferroptosis inducers, further emphasizing their critical role in modulating ferroptosis.

FSP1 serves as a critical factor in ferroptosis resistance and its transcriptional level can be regulated by the transcription factors NRF2 (Koppula et al, 2022), CREB (Nguyen et al, 2020), PPARα (Venkatesh et al, 2020) and non-coding RNAs such as RNA MEG3 and miR-214 (Fan et al, 2017). Furthermore, in colorectal cancer, the N-acetyltransferase NAT10 inhibits ferroptosis by N4-acetylating FSP1 mRNA to increase its mRNA stability and improve its translational efficiency (Zheng et al, 2022). In contrast, in acute lymphoblastic leukemia, the promoter of FSP1 is hypermethylated to silence its expression and evolve a selective dependence on GSH-centered anti-ferroptosis defenses (Pontel et al, 2022). In addition, recent findings have shed light on the

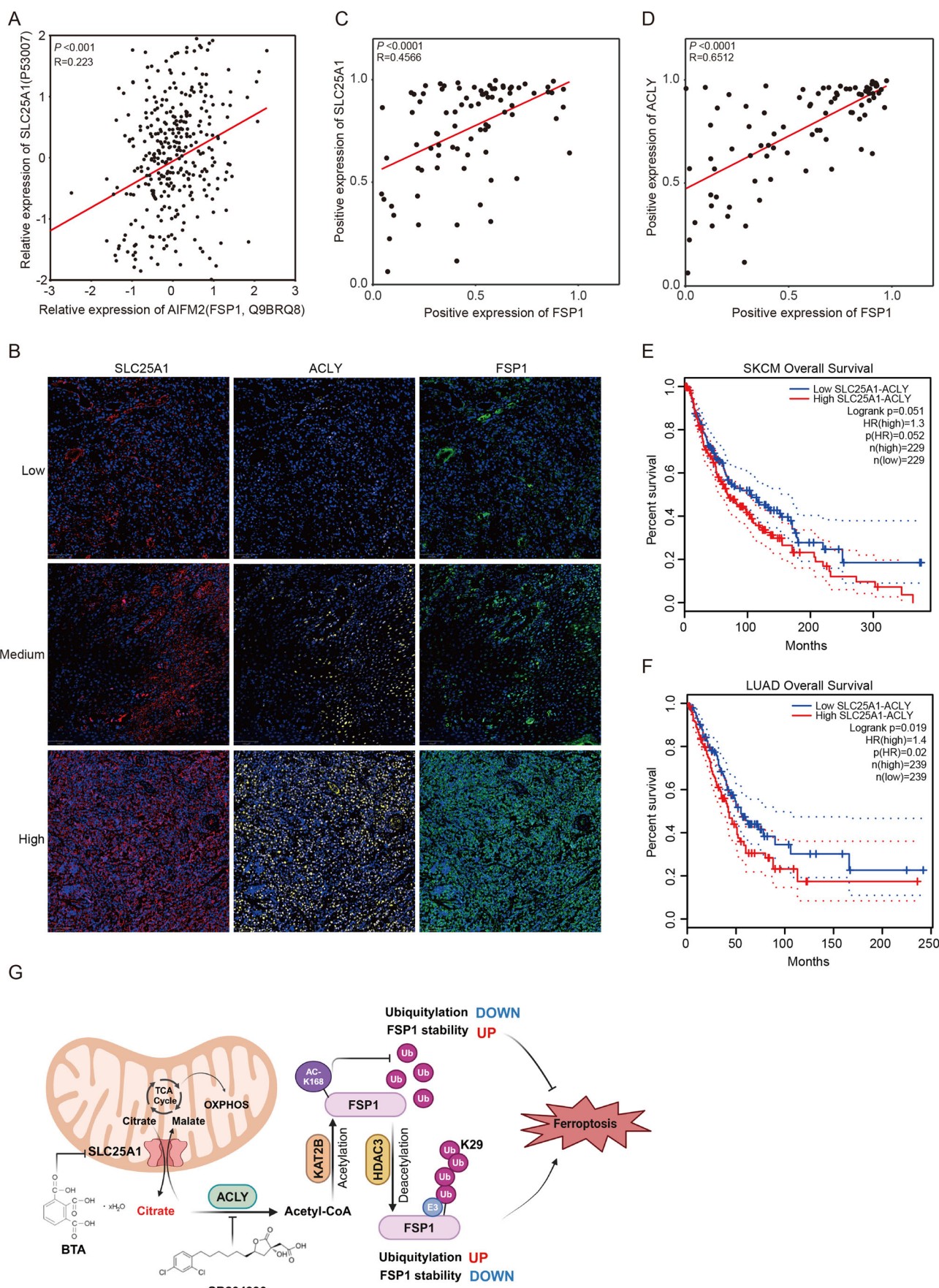

**Figure 7. The high expression SLC25A1-ACLY axis predicts the poor prognosis of cancer patients.**

(A) Relationship analysis of SLC25A1 and AIFM2 (FSP1) protein levels in the proteomics of cancer cell line encyclopedia were assessed by spearman correlation coefficient and log-rank tests, ***$p = 0.0002$, R = 0.223. (B) Multiplex immunofluorescence staining of SLC25A1, ACLY and FSP1 in a melanoma tissue microarray. Representative images showing SLC25A1 in red, ACLY in yellow and FSP1 in green. Scale bars: left, 100 μm. (C, D) Relationship analysis of SLC25A1 (C) or ACLY (D) and FSP1 positive expression in melanoma tissue microarray were assessed by spearman correlation coefficient and log-rank tests, ****$p = 4.83E{-}05$, R = 0.4566 (C); ****$p = 5.64E{-}06$, R = 0.6512 (D). (E, F) The overall survival (OS) in skin cutaneous melanoma (SKCM) (E) and lung adenocarcinoma (LUAD) (F) patients with different SLC25A1-ACLY mRNA levels were assessed by Kaplan–Meier survival curves and log-rank tests using GEPIA2. (G) A schematic model showing the mechanism by SLC25A1, in concert with ACLY, drives the export of citrate from mitochondria and the subsequent synthesis of acetyl-CoA in the cytosol, thereby facilitating FSP1 acetylation at K168 site and preventing its proteasomal degradation. The blockade of SLC25A1-ACLY increases cell susceptibility to ferroptosis and markedly enhances ferroptosis-induced tumor suppression. Source data are available online for this figure.

intricate regulation of FSP1 activity and protein stability through ubiquitination mechanisms. It was reported that the E3 ubiquitin ligase TRIM21 (Gong et al, 2023) interacts with FSP1, facilitating K63 ubiquitination at residues K322 and K366 via K63 linkage. This modification promotes the translocation of FSP1 to the membrane, bolstering its anti-ferroptotic capabilities while leaving its protein stability unaffected. Moreover, the binding of the long non-coding RNA lncFAL to FSP1 leads to elevated FSP1 protein levels by outcompeting FSP1 for interaction with the E3 ligase TRIM69 (Yuan et al, 2022). This interference disrupts the TRIM69-mediated degradation of FSP1 protein in hepatocellular carcinoma. In cases of acute kidney injury, CD36 binds to FSP1, influencing its ubiquitination at K16 and K24 sites, leading to FSP1 degradation and induction of ferroptosis, thereby worsening kidney injury (Ma et al, 2024). Lysine acetylation can compete with ubiquitination to affect protein stability and cellular sub-localization. However, whether FSP1 is regulated by acetylation has not been further investigated. Our study reveals that FSP1 can undergo acetylation at the K168 residue, and that the acetyltransferase KAT2B resists K29-linked FSP1 ubiquitination, which is crucial for maintaining FSP1 stability. Notably, SLC25A1 and ACLY facilitate acetyl-CoA production, which in turn enhances FSP1 acetylation and protects against ferroptosis.

In summary, we report that the SLC25A1-ACLY axis promotes the acetylation of FSP1, thereby increasing FSP1 protein stability and inhibiting ferroptosis sensitivity. Furthermore, we reveal that FSP1 is acetylated on K168 residue by KAT2B acetyltransferase, which is counteracted by HDAC3 deacetylase. Pharmacological targeting of SLC25A1 and ACLY sensitizes cancer cells to ferroptosis and inhibits tumor growth both in vitro and in vivo (Fig. 7G), suggesting that an in-depth study of this pathway may provide a new strategy for ferroptosis-targeted cancer treatment.

## Methods

### Reagents and tools table

| Reagent/resource | Reference or source | Identifier or catalog number |
| --- | --- | --- |
| RSL3 | Selleck | S8155 |
| Erastin | Selleck | S7243 |
| Ferrostatin-1 | Selleck | S7242 |
| Z-VAD-FMK | Selleck | S7023 |

| Reagent/resource | Reference or source | Identifier or catalog number |
| --- | --- | --- |
| Necrostatin-1 | Selleck | S8037 |
| Bafilomycin A1 | Selleck | S1413 |
| BTA | Macklin | B801861 |
| SB204990 | Targetmol | T16861 |
| Dimethyl citrate | MCE | HY-N9542 |
| Sodium acetate | Targetmol | T23377 |
| IKE | Selleck | S8877 |
| Liproxstatin-1 | Selleck | S7699 |
| MG132 | MCE | HY-13259 |
| Cycloheximide | MCE | HY-12320 |
| PVDF membrane | Millipore | IPVH00010 |
| BSA | Beyotime | ST025 |
| FBS | Excell | FSP100 |
| Matrigel | Corning | 354234 |
| PEI | Polysciences | 23966-1 |
| Polybrene | Sigma-Aldrich | 638132 |
| Puromycin | InvivoGen | ant-pr-1 |
| DAPI | Sigma-Aldrich | D9542 |
| Protein A/G Magnetic Beads | Thermo Fisher | 88802 |
| Anti-DYKDDDDK Magnetic Agarose | Thermo Fisher | A36798 |
| **Experimental models** | | |
| Human cell: A375 | ATCC | CRL-1619 |
| Human cell: SK-MEL-28 | ATCC | HTB-72 |
| Human cell: SK-MEL-30 | Dr. Jun Wan' laboratory | N/A |
| Human cell: A549 | ATCC | CCL-185 |
| Human cell: HT29 | ATCC | HTB-38 |
| Human cell: RKO | ATCC | CRL-2577 |
| Human cell: Huh-7 | ATCC | AC100423 |
| Human cell: HepG2 | ATCC | HB-8065 |
| Human cell: HT1080 | ATCC | CCL-121 |
| Human: HEK293T | ATCC | CRL-11268 |
| Mouse: BALB/c-nude | Vital River | N/A |
| Mouse: C57BL/6J | Vital River | N/A |

| Reagent/resource | Reference or source | Identifier or catalog number |
|---|---|---|
| **Recombinant DNA** | | |
| *pCDH-Flag-SLC25A1* | This paper | N/A |
| *PLV[EXP]-FSP1-Flag* | This paper | N/A |
| *PLV[EXP]-FSP1-Flag-K88R* | This paper | N/A |
| *PLV[EXP]-FSP1-Flag-K168R* | This paper | N/A |
| *PLV[EXP]-FSP1-Flag-K182R* | This paper | N/A |
| *PLV[EXP]-FSP1-Flag-K193R* | This paper | N/A |
| *PLV[EXP]-FSP1-Flag-K199R* | This paper | N/A |
| *PLV[EXP]-FSP1-Flag-K225R* | This paper | N/A |
| *PLV[EXP]-FSP1-Flag-K230R* | This paper | N/A |
| *PLV[EXP]-FSP1-Flag-K314R* | This paper | N/A |
| *PLV[EXP]-FSP1-Flag-K366R* | This paper | N/A |
| *PLV[EXP]-FSP1-Flag-G2A* | This paper | N/A |
| *pLX304-V5-HDAC3* | This paper | N/A |
| *pLX304-V5-ACLY* | This paper | N/A |
| *pCMV3-Myc-KAT2B* | This paper | N/A |
| *pcDNA3.1-HA-Ub* | This paper | N/A |
| *pcDNA3.1-HA-Ub K11R* | This paper | N/A |
| *pcDNA3.1-HA-Ub K29R* | This paper | N/A |
| *pcDNA3.1-HA-Ub K48R* | This paper | N/A |
| *pcDNA3.1-HA-Ub K29R only* | This paper | N/A |
| **Antibodies** | | |
| *Rabbit anti-SLC25A1* | Proteintech | 15235-1-AP |
| *Rabbit anti-GAPDH* | CST | 2118S |
| *Rabbit anti-ACLY* | Proteintech | 15421-1-AP |
| *Rabbit anti-SLC7A11* | Proteintech | 26864-1-AP |
| *Rabbit anti-GCH1* | Proteintech | 28501-1-AP |
| *Rabbit anti-GPX4* | Proteintech | 30388-1-AP |
| *Rabbit anti-FSP1* | Proteintech | 20886-1-AP |
| *Rabbit anti-ACSL4* | Proteintech | 22401-1-AP |
| *Rabbit anti-ACSS1* | Proteintech | 17138-1-AP |
| *Rabbit anti-ACSS2* | Proteintech | 16087-1-AP |
| *Rabbit anti-PDHA1* | Proteintech | 18068-1-AP |
| *Mouse anti-pan Acetylation* | Proteintech | 66289-1-Ig |
| *Rabbit anti-Myc tag* | CST | 2278S |
| *Rabbit anti-V5 tag* | CST | 13202S |
| *Rabbit anti-Flag tag* | CST | 14793S |
| *Rabbit anti-HA tag* | CST | 3724S |
| *Rabbit anti-HDAC3* | Proteintech | 10255-1-AP |
| *Rabbit anti-KAT2B* | Proteintech | 28770-1-AP |
| *Rabbit anti-Ubiquitin* | CST | 20326S |
| *Anti-rabbit IgG, HRP-linked* | CST | 7074S |
| *Anti-mouse IgG, HRP-linked* | CST | 7076S |
| *Rabbit mAb IgG Isotype Control* | CST | 3900S |

| Reagent/resource | Reference or source | Identifier or catalog number |
|---|---|---|
| **Oligonucleotides and other sequence-based reagents** | | |
| *shRNAs sequences* | This study | Table 1 |
| *sgRNA sequences* | This study | Table 2 |
| *Primers of real-time qPCR* | This study | Table 3 |
| **Software** | | |
| GraphPad Prism 8 | https://www.graphpad.com | N/A |
| Image J | https://imagej.en.softonic.com | N/A |
| QuPath | https://github.com/qupath/ | |
| **Other** | | |
| *CellTiter-Glo Luminescent Cell Viability Assay* | Promega | G7572 |
| *SYTOX Green nucleic acid stain* | Invitrogen | S7020 |
| *BODIPY 581/591 C11* | Invitrogen | D3861 |
| *GeneJET RNA Purification kit* | Thermo scientific | K0732 |
| *2 x SYBR Green qPCR Master Mix* | Selleck | B21202 |
| *Citrate Assay Kit* | Solarbio | BC2150 |
| *acetyl-CoA Assay Kit* | Sigma-Aldrich | MAK039 |
| *Cell Mitochondria Isolation Kit* | Beyotime | C3601 |
| *Plasma Membrane Protein Isolation Kit* | Invent Biotechnologies | SM005 |
| *NADP$^+$/NADPH Assay Kit* | Beyotime | S0179 |
| *GSH and GSSG Assay Kit* | Beyotime | S0053 |
| *ALT Assay Kit* | Nanjing Jiancheng | C009-2-1 |
| *AST Assay Kit* | Nanjing Jiancheng | C010-2-1 |
| *Human CoQ10 ELISA Kit* | X-Y Biotechnology | XY9H4197 |
| *Opal 6-Plex Detection Kit* | Akoya Biosciences | NEL811001KT |

## Cell culture

The human melanoma cell lines A375, SK-MEL-28, human lung cell line A549, human colorectal cancer cell lines HT29, RKO, human hepatocellular carcinoma cell line Huh-7, HepG2, human fibrosarcoma cell line HT1080 and human embryonic kidney cell line HEK293T were obtained from the American Type Culture Collection (ATCC), and human melanoma cell line SK-MEL-30 was given from Dr. Jun Wan' laboratory in Shenzhen PKU-HKUST Medical Centre in China. All cell lines were cultured in DMEM medium (Gibco, C11995500BT) supplemented with 10% fetal bovine serum (FBS) (Excell, FSP100) at 37 °C incubator with 5% $CO_2$.

All cells were cultured in a 10 cm plate and subcultured into 96-well- or 6-well plates for cell death, cell viability and lipid-peroxidation measurements. The cells were treated with reagents including the ferroptosis inducers RSL3 (Selleck, S8155) and erastin (Selleck, S7242), ferroptosis inhibitors Ferrostatin-1 (Fer-1)

**Table 1. shRNA sequences in this study.**

| Gene name/Species | shRNA sequences |
|---|---|
| SLC25A1-Homo | 1. CCATCCGCTTCTTCGTCATGA<br>2. CAGAGTGTCCTGCTACCTTTG |
| Slc25a1-Mus | 1. CTGCGGCTTGAAGATCCTAAA<br>2. GTATTCATCATCTACGATGAA<br>3. CATCGAAATCTGCATCACCTT |
| ACLY-Homo | 1. GCCTCAAGATACTATACATTT<br>2. CCTATGACTATGCCAAGACTA |
| KAT2B-Homo | 1. GCAGACTTACAGCGAGTCTTT<br>2. GCAGATACCAAACAAGTTTAT |
| HDAC3-Homo | 1. CCTTCCACAAATACGGAAATT<br>2. CAAGAGTCTTAATGCCTTCAA |
| ACSS1-Homo | 1. CTGTTGCTGAAATACGGTGAT<br>2. CAAGGTGGTTATCACCTTCAA |
| ACSS2-Homo | 1. GCTTGGAGATAAAGTTGCTTT<br>2. CGGTTCTGCTACTTTCCCATT |
| PDHA1-Homo | 1. GCCAATCAGTGGATCAAGTTT<br>2. CGAATGGAGTTGAAAGCAGAT |

(Selleck, S7243), apoptosis inhibitor Z-VAD-FMK (Z-V) (Selleck, S7023), necrosis inhibitor Necrostatin-1 (Nec-1) (Selleck, S8037), autophagy inhibitor Bafilomycin A1 (Baf-A1) (Selleck, S1413), SLC25A1 inhibitor BTA (Macklin, B801861), ACLY inhibitor SB204990 (Targetmol, T16861), dimethyl Citrate (MCE, HY-N9542), and sodium acetate (Targetmol, T23377).

## Human CRISPR SLCs knockout screen and data analysis

The screening utilized the human SLC family CRISPR knockout library (Addgene, Pooled Library #132552). Briefly, $2 \times 10^8$ A375 cells were transduced with a pooled human SLC lentiviral sgRNA library at a multiplicity of infection (MOI) of 0.3 for 24 h, followed by selection with 2 μg/ml puromycin (InvivoGen, ant-pr-1) for 7 days. After selection, $9 \times 10^7$ cells were treated with or without 2.5 μM RSL3. On day 8, $1 \times 10^7$ cells were harvested from the surviving population of RSL3-treated cells and untreated samples. Genomic DNA was extracted using Blood & Cell Culture DNA Kits (Qiagen, 13343) and sgRNA sequences were amplified by PCR using NEBNext® Q5® Hot Start HiFi PCR Master Mix (NEB, M0543L). The primers used were SLC_ArrayF: TAA CTT GAA AGT ATT TCG ATT TCT TGG CTT TAT ATA TCT TGT GGA AAG GAC GAA ACA CCG, SLC_ArrayR: ACT TTT TCA AGT TGA TAA CGG ACT AGC CTT ATT TTA ACT TGC TAT TTC T. Subsequently, deep sequencing was performed at Novogene Company to assess the relative enrichment or dropout of different sgRNA sequences.

## Lentiviral-mediated gene knockdown

Lentiviruses were produced through co-transfecting lentiviral plasmid, psPAX2 packaging plasmids and pMD2.G envelope expressing plasmid into HEK293T cells by PEI (Polysciences, 23966-1). The supernatants containing lentiviruses were harvested 48 h and 72 h post-transfection and passed through a 0.45-mm filter. A375, SK-MEL-28, SK-MEL-30, HT29, RKO, Huh-7, HepG2 and A549 cells were infected with lentiviral for 24 h, followed by

**Table 2. sgRNA sequences in this study.**

| Gene name/Species | sgRNA sequences |
|---|---|
| FSP1-Homo | AACTCGGGAAGACGACCAAG |

selecting with fresh medium containing 2 μg/mL puromycin (InvivoGen, ant-pr-1) to obtain stably infected cells. The shRNA sequences used were listed in Table 1.

## Generation of FSP1-knockout cell lines

Knockout of FSP1 in A375 and A549 cells was performed using sgRNAs and CRISPR/Cas9 technology. sgRNAs were cloned into the lentiviral lentiCRISPR V2 vector, and then lentiviruses were obtained by HEK293T cells. A375 and A549 cells were infected with lentivirus for 24 h, followed by selecting with 2 μg/ml puromycin. After 3 days, the stably cells were subjected to clonal isolation into 96-well plates and expanded through passage. After 3–4 weeks, each colony was confirmed by immunoblot to confirm FSP1 deletion. The FSP1 sgRNA sequences used were listed in Table 2.

## Cell-death assays

Cell death was detected using SYTOX Green nucleic acid stain (Invitrogen, S7020). In brief, cells were seeded into 6-well plates at a density of $4 \times 10^5$/well. The next day, the cells were subjected to indicated treatment, and then incubated with 167 nM SYTOX Green at 37 °C in the dark. After 30 min, staining solution was removed, and the cells were washed twice with phosphate-free buffer to eliminate any unbound probe. Subsequently, the cell images were randomly captured by a fluorescence microscopy with excitation at 504 nm and emission at 523 nm. The percentage of SYTOX Green-positive cells was analyzed as SYTOX Green-positive cell number/total cell number from three fields.

## Cell viability assays

Cell viability was measured using CellTiter-Glo Luminescent Cell Viability Assay (Promega, G7572) according to manufacturer's protocol. Briefly, cells were seeded into 96-well plates at appropriate density and then incubated with designated treatments as described in the separate experiments. Subsequently, each well was added with 100 μl CellTiter-Glo Reagent and subjected at room temperature for 10 min on a shaker. Finally, the luminescence was detected with a microplate reader (BIOTEK, Vermont, USA).

## Measurement of lipid peroxidation

Cells were seeded into 6-well plates at a density of $2 \times 10^5$/well. On the second day, the cells were subjected to indicated treatment, and then collected, stained with 200 μl PBS containing 5 μM BODIPY 581/591 C11(Invitrogen, D3861) at 37 °C in the dark for 30 min. Subsequently, lipid peroxidation was evaluated through flow cytometer (BD Biosciences, CA, USA). In the experiments, at least $1 \times 10^4$ cells were measured in each group, and repeated three times.

**Table 3. Real-time qPCR primer sequences in this study.**

| Gene name/Species | sequences 5-3' |
|---|---|
| SLC25A1-Homo | Forward Primer: GACAACGGGGTGAGGGC<br>Reverse primer: ATAGCTCCGAAGACCCCAGT |
| Slc25a1-Mus | Forward Primer: AGCTCCTTGCTCTACGGCT<br>Reverse Primer: ACCGCACAATAGTCCTCTCCT |
| PTGS2-Homo | Forward Primer: CTGGCGCTCAGCCATACAG<br>Reverse Primer: CGCACTTATACTGGTCAAATCCC |
| Ptgs2-Mus | Forward Primer: TTCCAATCCATGTCAAAACCGT<br>Reverse Primer: AGTCCGGGTACAGTCACACTT |
| Chac1-Mus | Forward Primer: CTGTGGATTTTCGGGTACGG<br>Reverse Primer: CTCGGCCAGGCATCTTGTC |
| CS-Homo | Forward Primer: TGCTTCCTCCACGAATTTGAAA<br>Reverse Primer: CCACCATACATCATGTCCACAG" |
| ACO2-Homo | Forward Primer: CCCTACAGCCTACTGGTGACT<br>Reverse Primer: TGTACTCGTTGGGCTCAAAGT |
| ACLY-Homo | Forward Primer: ATCGGTTCAAGTATGCTCGGG"<br>Reverse Primer: GACCAAGTTTTCCACGACGTT |
| FSP1-Homo | Forward Primer: AGACAGGGTTCGCCAAAAAGA<br>Reverse Primer: CAGGTCTATCCCCACTACTAGC |
| GAPDH-Homo | Forward Primer: ACAACTTTGGTATCGTGGAAGG<br>Reverse Primer: GCCATCACGCCACAGTTTC |

## Mitochondrial fractionation

Cell mitochondria were collected using the cell mitochondria isolation kit (Beyotime, C3601) (Bai et al, 2022; Tao et al, 2017). In brief, A375 and A549 cells were resuspended in the provided mitochondria extraction reagent and homogenized with a micro-homogenizer, then incubated on ice for 15 min. The homogenate was centrifuged at $1000 \times g$ for 10 min at 4 °C, and the supernatant was collected for further centrifugation at $11,000 \times g$ for another 10 min at 4 °C. The mitochondrial fraction was isolated from the pellet, and the cytosol and nucleus fraction was obtained from the supernatant.

## Biochemical assay

The concentrations of citrate (Solarbio, BC2150) of cytoplasm and mitochondrion, acetyl-CoA (Sigma-Aldrich, MAK039), NADPH (Beyotime, S0179), GSH (Beyotime, S0053), Coenzyme Q10 (CoQ10, X-Y Biotechnology, XY9H4197) or ALT (Nanjing Jiancheng, C009-2-1), AST (Nanjing Jiancheng, C010-2-1) of serum in indicated cell samples were measured using the kit according to the manufacturer's instructions. All experiments were repeated three times.

## qRT-PCR analysis

Total RNA was extracted from cells following the manufacturer's protocol for the GeneJET RNA Purification kit (Thermo scientific, K0732). The mRNA was reversely transcribed to generate cDNA using the HiScript® II Q RT SuperMix for qPCR (+gDNA wiper) kit (Vazyme, R223-01). qRT-PCR was performed by 2 x SYBR Green qPCR Master Mix (Selleck, B21202) with a Bio-Rad Multicolor Real-time PCR Detection System (iQTM5, Bio-Rad, CA, USA). The primers used were listed in Table 3, GAPDH was used as the internal control, and the relative expression of indicated genes was calculated using the 2-ΔΔCT method.

## Preparation of cell plasma membrane proteins

Cell plasma membrane proteins were extracted from cells using the Minute™ Plasma Membrane Protein Isolation and Cell Fractionation Kit (Invent Biotechnologies, SM-005) as follows. A375 cells were harvested from 150 mm culture dishes by scraping, followed by incubation in Buffer A on ice for 10 min. After incubation, the cell suspension was vortexed vigorously for 30 s. The suspension was rapidly transferred to a centrifuge tube column cannula and centrifuged at $16,000 \times g$ for 30 s at 4 °C. The centrifuge column was discarded, and the precipitate in the receiving tube was resuspended by vortexing for 10 s. This was followed by centrifugation at $700 \times g$ for 1 min at 4 °C. The resulting supernatant was collected and centrifuged at $16,000 \times g$ for 30 min at 4 °C to obtain the precipitate. Buffer B was added to the precipitate, which was vortexed to resuspend the pellet and subsequently centrifuged at $7800 \times g$ for 5 min at 4 °C. At this point, the cytoplasmic fraction (pellet) was preserved. The supernatant was collected, diluted with PBS, and centrifuged again at $16,000 \times g$ for 30 min at 4 °C. The final pellet obtained represented the plasma membrane fraction.

## Immunoblots and immunoprecipitation

Cells were lysed with RIPA buffer (Beyotime, P0013B) supplemented with protease inhibitor (Bimake, B14001) for 30 min on ice. Total protein concentration was determined using BCA protein assay kit (Beyotime, P0012). Next, protein was separated by SDS-PAGE gels, transferred to PVDF membranes (Millipore, IPVH00010), probed with indicated primary antibodies and subsequently with appropriate secondary antibodies. Finally, visualization of the target protein bands was performed by the ChemDocTM MP Imaging System (Bio-Rad, CA, USA). For immunoprecipitation, cells were lysed with NP40 buffer (Beyotime, P0013F) supplemented with protease inhibitor. Then, lysates were incubated with Flag magnetic beads (Thermo Fisher, A36798), or Protein A/G magnetic beads (Thermo Fisher, 88802). The immunoprecipitates were washed five times with NP40 buffer, and then separated by SDS-PAGE and immunoblotted with indicated antibodies.

## CHX chase assays

Cells were seeded into 6-well plates and incubated overnight. On the next day, cells were exposed to 60 μg/mL CHX (MCE, HY-12320) for various durations, ranging from 0 to 12 h. Subsequently, proteins were extracted and dictated to immunoblots. The FSP1 protein level was quantified by ImageJ software.

## Ubiquitination assays

For polyubiquitination analysis of FSP1 in HEK293T cells, the cells were transfected with plasmids expressing HA-ubiquitin (WT, K11R, K29R or K48R) and FSP1-Flag. 48 h after transfection, the cells were harvested in 500 μL SDS lysis buffer (1.5% SDS, 50 mM Tris-Cl, pH 6.8) and heated at 100 °C for 15 min. Subsequently, 100 μL of the cell lysate was used as the input, while the remaining lysate was diluted

tenfold with cold BSA buffer (0.5% BSA, 0.5% NP-40, 180 mM NaCl, 50 mM Tris-Cl, pH 6.8), and immunoprecipitated with the Flag magnetic beads for 4 h at 4 °C on a rotary shaker. The beads were collected and then washed 5 times with cold BSA buffer. The beads were boiled for 10 min with 100 μL 2 x SDS buffer and then analyzed by immunoblotting with anti-HA to detect FSP1 ubiquitination.

For polyubiquitination analysis of endogenous FSP1 in indicated cells. After 12 h of 10 μM MG132 (MCE, HY-13259) treatment, and the cells were harvested as mentioned above and the cell lysates were immunoprecipitated with the anti-FSP1 antibody for overnight at 4 °C. On the second day, Protein A/G magnetic beads were added to the cell lysate and incubated at 4 °C for 4 h, then analyzed by immunoblotting with anti-Ub to detect FSP1 ubiquitin.

## Immunofluorescence

HT1080 cells with overexpressing GFP-FSP1 WT/K168R/K168Q were fixed in 4% paraformaldehyde for 30 min at room temperature, and then permeabilized with 0.5% Triton X-100 for 5 min and blocked with 5% BSA for 1 h. After washing by PBS, nuclear was staining by DAPI (Sigma-Aldrich, D9542). Finally, the cells were observed on the LSM 900 Confocal Microscopy System (Zeiss, Oberkochen, GER) for image acquisition and data analysis.

## FENIX assays

Firstly, recombinant human FSP1 proteins (rhFSP1 WT, rhFSP1 K168Q) were produced in Escherichia coli, and purified through affinity chromatography using a Ni-NTA system as described previously. Next, the egg PC liposomes (1 mM, extruded to 100 nm) and STY-BODIPY (1 μM) and indicated concentration of liproxstatin-1 (Lip-1) (Selleck, S7699), K2, NADPH, NADPH + K2, NADPH + K2 + rhFSP1 WT, NADPH + K2 + rhFSP1 K168Q or vehicle (DMSO) were vortexed in PBS (10 mM, pH7.4). Then, DTUN (200 mM in EtOH) were added to the aliquots. Subsequently, 200 μl aliquots of liposomes were incubated in 96-well plates for 20 min at 37 °C. Finally, the plate was mixed for 5 min and kinetic data of STY-BODIPYOX was determined at 485 nm (λex) and 528 nm (λem) using Mithras LB940 microplate reader (Berthold Technologies, Bad Wildbad, GER).

## Animal studies

All experimental protocols for animal studies received approval by the Animal Research Ethics Committee of Shenzhen Bay Laboratory (Approved Protocol ID: AEYCQ202101). Male C57BL/6J and BALB/c nude mice were purchased from Vital River (Beijing, China). All mice were kept under specific-pathogen-free conditions with sufficient water and food. For animal studies, mice were randomized into separate cages. For animal studies, mice were randomly divided into separate cages.

## Xenograft model in mice

To investigate the effect of pharmacological inhibition of SLC25A1 and ACLY on ferroptosis in vivo, we constructed the melanoma cancer xenograft mouse model. $3 \times 10^6$ WT A375 cells were subcutaneously injected into the right flank of 5–6 weeks old BALB/c nude mice, and once the tumor volume reached 50-70 mm³, mice were randomly divided into six groups ($n = 7$): (1) Vehicle group; (2) IKE (50 mg/kg) group; (3) BTA (50 mg/kg) + IKE (50 mg/kg) group; (4) BTA (50 mg/kg) + IKE (50 mg/kg) + Lip-1 (10 mg/kg) group; (5) SB204990 (50 mg/kg) + IKE (50 mg/kg); and (6) SB204990 (50 mg/kg) + IKE (50 mg/kg) + Lip-1 (10 mg/kg) group. Mice were i.p. every two days for a total of 13 days, during which tumor size of the mice were measured, and tumor size was calculated by the following formula: $0.5 \times \text{length} \times \text{width}^2$.

## Liver ischemia–reperfusion injury (IRI) model in mice

The liver IRI model as described previously. Before undergoing IRI, 8–10 weeks old male C57BL/6J mice were received corresponding treatment. Next, the mice were anesthetized by i.p. 0.5% pentobarbital (200 μl), and then abdominal cavity of mice was opened, an atraumatic clamp was placed on left hepatic artery for 45 min. Subsequently, the clamp was removed and the liver was reperfused. Sham-operated mice received identical surgical procedures, except left hepatic artery clamping. After 24 h of reperfusion, all mice were sacrificed, and then blood and liver were collected for analysis.

## Immunohistochemistry

The tissues were fixed with 4% paraformaldehyde for 48 h and embedded in paraffin, then sectioned to 5–8 μm slides. After dewaxing, hydration and antigen retrieval, the slides were incubated with the Ki67, FSP1, SLC25A1, or MDA antibody overnight at 4 °C. Next day, after washed with PBST for 3 times, the slides were incubated with biotinylated secondary antibody for 1 h. Finally, the slides were developed by DAB.

## Multiplex immunofluorescence (mIF) staining

The melanoma tissue microarray (TMA) was purchased from Shanghai Outdo Biotech Co., Ltd (Shanghai, China). mIF staining was performed using the Opal 6-Plex Detection Kit (Akoya Biosciences, NEL811001KT) according to the manufacturer's instructions. Briefly, TMA slide was dewaxed with xylene and then rehydrated through a graded ethanol series before antigen repair using high pressure with Opal-AR6 Buffer for 30 min. After blocking with 1 × Antibody Diluent/Block for 10 min at room temperature, the staining process was performed in a serial manner 3 times, including incubation with a primary antibody (SLC25A1, 1:100; ACLY, 1:100 and FSP1, 1:100), a secondary antibody conjugated to horseradish peroxidase, and sequentially an Opal reactive fluorophore (SLC25A1, Opal 690; ACLY, Opal 570 and FSP1, Opal 520). Finally, the cell nuclei were counterstained with DAPI. The four-color Opal slide was visualized using Akoya Biosciences PhenoCycler-Fusion (Akoya Biosciences, MA, USA), followed by application of spectral unmixing to distinguish between the four different fluorescent signals. Unmixed images were analyzed for quantitative fluorescence intensity using QuPath software.

## RNA-seq and data analysis

Total RNA was extracted using TRIzol reagent (Invitrogen, CA, USA) to commercial RNA-seq analysis (Novogene, Beijing, China). The cDNA library was constructed through NEBNext® Ultra™ RNA Library Prep Kit for Illumina® (NEB, MA, USA), and then

sequenced by the Illumina NovaSeq 6000 (Lllumina, CA, USA). Next, the sequence quality was verified by fastp software and reference genome of Homo sapiens GRCh38 was read mapping through HISAT2. Subsequently, FPKM of each gene was calculated by featureCounts (v1.5.0-p3). Differential expression analysis was performed using the DESeq2 R package (1.20.0), $padj \leq 0.05$ and $|\log2(foldchange)| \geq 1$ were set as the threshold for significantly differential expression.

## Proteomics and analysis

Intracellular proteins were extracted through urea buffer (8 M in 25 mM ammonium bicarbonate) with sonication at 4 °C and quantified using the BCA assay. The disulfide bonds in proteins were reduced by DTT, and cysteines were alkylated by IAA, followed by protein digestion through trypsin. Peptide desalting was subsequently implemented by C18 mini tubes, and the eluents were collected and dried at 4 °C in a vacuum. Finally, total peptides were separated using Orbitrap Fusion Lumos + LC-MS/MS system (Thermo Scientific, MA, USA). For raw data analysis, proteome library was constructed using Proteome Discoverer software, and then differential expression analysis was performed using the DESeq2 R package (1.20.0), $padj \leq 0.05$ and $|\log2(foldchange)| \geq 1$ were set as the threshold for significantly differential expression.

## Lipidomics analyses

Lipidomics was performed by Shanghai Applied Protein Technology Co., Ltd (Shanghai, China). In brief, After the extraction of intracellular metabolites, reverse phase chromatography was selected for LC separation using CSH C18 column (1.7 μm, 2.1 mm × 100 mm, waters), and mass spectra was acquired by Q-Exactive Plus in positive and negative mode, respectively. Finally, lipid identification was achieved using Lipid Search Library, and both mass tolerance for precursor and fragment were set to 5 ppm.

## LC-MS/MS for the quantitative determination of myristic acid and palmitic acid

The cell pellet was suspended in 150 μl of PBS, followed by the addition of 300 μl of methanol and 600 μl of chloroform, and mixed for 5 min. The sample was centrifuged at 1000 × g for 5 min, and the lower lipid-rich layer was collected. The remaining aqueous phase was re-extracted by adding 150 μl of methanol and 300 μl of chloroform, and vortexing afterwards. The sample was centrifuged again at 1000 × g for 5 min. The collected lipid-rich layers were combined and evaporated to dryness under vacuum. The dried extract was reconstituted with 30 μl of methanol.

The liquid chromatography-tandem mass spectrometry (LC-MS/MS) analyses were performed using a QTRAP® 6500 + LC-MS/MS system (Sciex, Massachusetts, USA) equipped with a Waters Acquity UPLC® BEH C18 column (2.1 × 100 mm, 1.7 μm). Myristic acid and palmitic acid were analyzed in negative ion multiple reaction monitoring (MRM) mode. Mobile phase A consisted of 40% water and 60% acetonitrile with 10 mM ammonium acetate. Mobile phase B consisted of 90% isopropyl alcohol and 10% acetonitrile. The analysis was carried out with a flow rate of 0.3 ml/min using the following elution gradient: 0–2 min, 90–70% phase

A; 2–8 min, 70–0% phase A; 8–10 min, 0% phase A; 10–10.1 min, 0–90% phase A; 10.1–11.5 min, 90–0% phase A; and 11.5–13 min, 0–90% phase A. The column temperature was 50 °C. The autosampler temperature was 15 °C, and the injection volume was 1 μl. The electrospray ionization (ESI) source conditions were set as follows: ion source gas 1, 50 psi; ion source gas 2, 50 psi; capillary temperature, 500 °C; and spray voltage, 5 kV (positive) or −4.5 kV (negative). The following parent-to-daughter transitions were monitored: $m/z$ 273.2 $[M + HCOO]^-$ to $m/z$ 227.2 for myristic acid with a collision energy (CE) of −18 V and a declustering potential (DP) of −5 V, $m/z$ 255.1 $[M-H]^+$ to $m/z$ 237.3 for palmitic acid with CE of −30 V and DP of −170 V.

## Statistical analysis

All experiments were independently conducted at least three times with a similar outcome. Statistical analyses involved were performed by GraphPad Prism 10, and the results were all presented as the mean ± SD. Significant differences between the two groups were assessed using unpaired two-tailed Student's t test, and multiple comparisons were analyzed by one-way analysis of variance (ANOVA) or two-way ANOVA, the $p$-value less than 0.05 is indicated statistically significant. $P$ values are denoted as follows: $*p < 0.05$; $**p < 0.01$; $***p < 0.001$, $****p < 0.0001$.

# Data availability

The correlation between the SLC25A1 levels and FSP1 protein expression were derived from DepMap (https://depmap.org/portal/). Patient survival was obtained from GEPIA2 (http://gepia2.cancer-pku.cn/#index). The authors confirm that the data supporting the findings of this study are available within the article [and/or] its supplementary materials. The RNA-Seq data supporting the findings of this study have been deposited in Genome Sequence Archive (GSA) under accession number HRA007497 (https://ngdc.cncb.ac.cn/gsa-human/browse/HRA007497), and the proteomics data supporting the findings of this study have been deposited in ProteomeXchange Consortium (https://www.ebi.ac.uk/pride/login, Project Accession: PXD052299, Reviewer Token: aTyDr4JiFkZu).

The source data of this paper are collected in the following database record: biostudies:S-SCDT-10_1038-S44318-025-00369-5.

# Peer review information

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

## Acknowledgements

This work was supported by Shenzhen Medical Research Fund (B2302018), National Key R&D Program of China (2022YFA0912600 and 2023YFC2308200), National Natural Science Foundation of China (32100579, 32370800, 82341011, and 82204192), Major Program (S201101004) and Open Fund Program (SZBL2021080601004) of Shenzhen Bay Laboratory. The authors are grateful to Biochemical Analysis Core, Multi-omics Mass Spectrometry Core, Bio-Imaging Core and Animal Laboratory Center of Shenzhen Bay Laboratory for help with technical assistance.

## Author contributions

**Wei Li**: Data curation; Investigation; Writing—original draft; Writing—review and editing. **Jing Han**: Investigation; Writing—review and editing. **Bin Huang**: Investigation; Writing—review and editing. **Tengteng Xu**: Investigation; Writing—review and editing. **Yihong Wan**: Investigation. **Dan Luo**: Investigation. **Weiyao Kong**: Investigation. **Ying Yu**: Investigation. **Lei Zhang**: Writing—review and editing. **Yong Nian**: Conceptualization; Funding acquisition; Writing—review and editing. **Bo Chu**: Conceptualization; Funding acquisition; Project administration; Writing—review and editing. **Chengqian Yin**: Conceptualization; Data curation; Funding acquisition; Writing—original draft; Project administration; Writing—review and editing.

Source data underlying figure panels in this paper may have individual authorship assigned. Where available, figure panel/source data authorship is listed in the following database record: biostudies:S-SCDT-10_1038-S44318-025-00369-5.

## Disclosure and competing interests statement

The authors declare no competing interests.

