## [Peer Review File · The EMBO Journal]

Cytosolic citrate is maintained by SLC25A1 and ACLY and governs ferroptosis susceptibility of cancer cells via FSP1 acetylation

Wei Li, Jing Han, Bin Huang, Tengting Xu, Yihong Wan, Dan Luo, Weiyao Kong, Ying Yu, Lei Zhang, Yong Nian, Bo Chu, and Chengqian Yin

Corresponding author(s): Chengqian Yin (yincq@szbl.ac.cn) , Yong Nian (ynian@njucm.edu.cn), Bo Chu (chubo123@sdu.edu.cn)

Review Timeline:

Submission Date:	29th May 24
Editorial Decision:	24th Jul 24
Revision Received:	4th Oct 24
Editorial Decision:	8th Nov 24
Revision Received:	13th Dec 24
Accepted:	7th Jan 25

Editor: William Teale

Transaction Report:

Dear Prof. Yin,

Thank you again for the submission of your manuscript entitled "SLC25A1 orchestrates with ACLY to govern ferroptosis susceptibility via FSP1 acetylation" (EMBOJ-2024-118028) and for your patience during the unusually long review process. We have now received the reports from the referees, which I copy below.

As you can see from the comments, referee #1 expresses concern over the mechanistic depth to which your manuscript links FSP1 acetylation, stating that the regulation of ferroptosis may be better explained by alternative mechanisms. These are important concerns that will need to be addressed before your manuscript can be published in The EMBO Journal.

However, based on the overall interest expressed in the reports, I would like to discuss with you the feasibility of addressing these concerns in a revised version of the manuscript. I should add that it is The EMBO Journal policy to allow only a single major round of revision and that it is therefore important to resolve the main concerns at this stage. I am always available over Zoom; just let me know a convenient afternoon/evening for you. In case we decide to proceed, I include instructions preparing your manuscript for a potential resubmission below.

I would also like to point out that as a matter of policy, competing manuscripts published during this period will not be taken into consideration in our assessment of the novelty presented by your study ("scooping" protection). We have extended this 'scooping protection policy' beyond the usual 3 month revision timeline to cover the period required for a full revision to address the essential experimental issues. Please contact me if you see a paper with related content published elsewhere to discuss the appropriate course of action.

If invited to prepare a letter of response to the referees' comments, please bear in mind that this will form part of the Review Process File, and will therefore be available online to the community. For more details on our Transparent Editorial Process, please visit our website: <https://www.embopress.org/page/journal/14602075/authorguide#transparentprocess>

Again, please contact me at any time during revision if you need any help or have further questions.

Thank you very much again for the opportunity to consider your work for publication.

Best regards,

William

William Teale, Ph.D.
Editor
The EMBO Journal

When submitting your revised manuscript, please carefully review the instructions below and include the following items:

- 1) a .docx formatted version of the manuscript text (including legends for main figures, EV figures and tables). Please make sure that the changes are highlighted to be clearly visible.
- 2) individual production quality figure files as .eps, .tif, .jpg (one file per figure).
- 3) a .docx formatted letter INCLUDING the reviewers' reports and your detailed point-by-point response to their comments. As part of the EMBO Press transparent editorial process, the point-by-point response is part of the Review Process File (RPF), which will be published alongside your paper.
- 4) a complete author checklist, which you can download from our author guidelines ([https://wol-prod-cdn.literatumonline.com/pb-assets/embo-site/Author Checklist%20-%20EMBO%20J-1561436015657.xlsx](https://wol-prod-cdn.literatumonline.com/pb-assets/embo-site/Author%20Checklist%20-%20EMBO%20J-1561436015657.xlsx)). Please insert information in the checklist that is also reflected in the manuscript. The completed author checklist will also be part of the RPF.
- 5) Please note that all corresponding authors are required to supply an ORCID ID for their name upon submission of a revised manuscript.
- 6) We require a 'Data Availability' section after the Materials and Methods. Before submitting your revision, primary datasets produced in this study need to be deposited in an appropriate public database, and the accession numbers and database listed under 'Data Availability'. Please remember to provide a reviewer password if the datasets are not yet public (see <https://www.embopress.org/page/journal/14602075/authorguide#datadeposition>). If no data deposition in external databases is

needed for this paper, please then state in this section: This study includes no data deposited in external repositories. Note that the Data Availability Section is restricted to new primary data that are part of this study.

Note - All links should resolve to a page where the data can be accessed.

8) For data quantification: please specify the name of the statistical test used to generate error bars and P values, the number (n) of independent experiments (specify technical or biological replicates) underlying each data point and the test used to calculate p-values in each figure legend. The figure legends should contain a basic description of n, P and the test applied. Graphs must include a description of the bars and the error bars (s.d., s.e.m.).

9) We would also encourage you to include the source data for figure panels that show essential data. Numerical data can be provided as individual .xls or .csv files (including a tab describing the data). For 'blots' or microscopy, uncropped images should be submitted (using a zip archive or a single pdf per main figure if multiple images need to be supplied for one panel). Additional information on source data and instruction on how to label the files are available at .

10) We replaced Supplementary Information with Expanded View (EV) Figures and Tables that are collapsible/expandable online (see examples in <https://www.embopress.org/doi/10.15252/embj.201695874>). A maximum of 5 EV Figures can be typeset. EV Figures should be cited as 'Figure EV1, Figure EV2" etc. in the text and their respective legends should be included in the main text after the legends of regular figures.

12) Our journal encourages inclusion of *data citations in the reference list* to directly cite datasets that were re-used and obtained from public databases. Data citations in the article text are distinct from normal bibliographical citations and should directly link to the database records from which the data can be accessed. In the main text, data citations are formatted as follows: "Data ref: Smith et al, 2001" or "Data ref: NCBI Sequence Read Archive PRJNA342805, 2017". In the Reference list, data citations must be labeled with "[DATASET]". A data reference must provide the database name, accession number/identifiers and a resolvable link to the landing page from which the data can be accessed at the end of the reference. Further instructions are available at .

Further instructions for preparing your revised manuscript:

- a point-by-point response to the referees' comments, with a detailed description of the changes made (as a word file).
- a word file of the manuscript text
- individual production quality figure files (one file per figure)
- a complete author checklist, which you can download from our author guidelines (<https://www.embopress.org/page/journal/14602075/authorguide>).
- Expanded View files (replacing Supplementary Information)

The revision must be submitted online within 90 days; please click on the link below to submit the revision online before 22nd Oct 2024.

Referee #1:

Ferroptosis is a form of programmed cell death dependent on iron and characterized by the accumulation of lipid hydroperoxides, making it a potential target for cancer therapy. In the present study the authors study the role of members of the solute carrier (SLC) family of proteins in ferroptosis. Specifically, the study identifies SLC25A1 as a regulator of ferroptosis through CRISPR-Cas9 screening. SLC25A1, along with ACLY, drives the export of citrate from the mitochondria and its conversion to acetyl-CoA in the cytosol. The authors posit that the cytosolic pool of acetyl-CoA is crucial for promoting the acetylation and stabilization of FSP1, an anti-ferroptotic protein. The acetylation of FSP1 at the K168 residue is mediated by KAT2B and HDAC3. Inhibiting SLC25A1 and ACLY pharmacologically increases cancer cells' susceptibility to ferroptosis, suggesting that targeting the SLC25A1-ACLY pathway could be an effective strategy to sensitize cells to ferroptosis. My comment are limited to the molecular mechanism regulating ferroptosis which is my area of expertise. Overall, the study is of interest and provides insights into the biology of FSP1. Nevertheless, the acetylation of FSP1 and its functional implication in regulating ferroptosis appear less convincing. Some aspects of the study could be better explained by alternative mechanisms. Therefore, I would encourage the authors to consider this in order to provide a better mechanistic interpretation.

Major point:

- The connection between SLC25A1 and ACLY in ferroptosis is convincingly demonstrated. Nonetheless an alternative explanation that is independent on the acetylation of FSP1 could explain the sensitization observed. Loss of ACLY is expected to inhibit lipogenesis as the authors rightly acknowledge, can the authors exclude that the loss of ACLY is not promoting a shift in the MUFA/PUFA ratio. The analysis presented in figure EV4 could be more informative if this comparison is presented.
- Following up on my previous point where lack of ACLY could affect the availability of saturated and monounsaturated fatty acids, is there any change in the subcellular distribution of FSP1? If myristic acid, which is required for FSP1 myristoylation, is limiting, than a pool of non-membrane bound FSP1 could result in an increase in ferroptosis sensitivity. If that's the case, one should consider investigating the half-life of wt and non-myristoylated (G2A) FSP1.
- An important experiment the authors might wish to perform in order to provide support for the hypothesis would be to express the non-acetylatable and the acetyl-mimic mutant of FSP1 in FSP1-KO, ACLY/FSP1-2KO and SLC25A1/FSP1-2KO cells and assess their response to RSL3. This should provide unequivocal proof that SLC25A1 and ACLY are increasing ferroptosis sensitivity in an FSP1 dependent manner.
- Mass spectrometry analysis of immunoprecipitated FSP1 validating its acetylation beyond the use of antibodies its advisable.
- The authors use PTGS2 as a read out of "ferroptosis" this is probably a very unspecific approach and should be avoided since multiple stress response pathways can lead to PTGS2 upregulation. I would encourage to validate some of the finding using BODIPY-C11, which is a probe sensitive to alterations in membrane redox state.

Minor point:

- Line 57, PMID: 38297129 needs to be included
- Correct lipid peroxides to lipid hydroperoxides throughout the text

Referee #2:

The work is great. However, some issues remain that should be addressed to better support the concept.

1. Spaces are missing before inserted references in the text.
2. In figure 1F, efficiency of SLC25A1 overexpression in cells with SLC25A1 knockdown should be shown.
3. The full name of Fer-1 should be added in line 104 instead of line 111.
4. The authors state that SLC25A1 knockdown directly induced cell ferroptosis. But why did MDM staining show no difference

between sample with vehicle + sham and sample with Slc25a1 KD in figure 1N.

5. As the authors focused on liver IRI in vivo, why lung cancer cell line A549 was chosen in vitro experiments ?

6. Noting that figure 2A showed citrate levels in the total cell, cytoplasm, and mitochondria, the protocols for the isolation of cytoplasm and mitochondria should be described in methods. Additionally, fraction contamination should be ruled out by immunoblotting of markers for cytoplasm, nuclear and mitochondria.

7. The effect of inhibiting acetyl-CoA production by dimethyl citrate and sodium acetate should be demonstrated to support the result of Figure EV3J-S3K.

8. The IP result in figure 3H is flawed as the levels of K168R-FLAG in WCL are lower than others.

9. Bands of GAPDH in WCL are missing in Figure 4C, 4K, 5A, 5B, EV6A, EV6B, EV6H.

Point-by-point response to the reviewers' comments

Referee #1

Ferroptosis is a form of programmed cell death dependent on iron and characterized by the accumulation of lipid hydroperoxides, making it a potential target for cancer therapy. In the present study the authors study the role of members of the solute carrier (SLC) family of proteins in ferroptosis. Specifically, the study identifies SLC25A1 as a regulator of ferroptosis through CRISPR-Cas9 screening. SLC25A1, along with ACLY, drives the export of citrate from the mitochondria and its conversion to acetyl-CoA in the cytosol. The authors posit that the cytosolic pool of acetyl-CoA is crucial for promoting the acetylation and stabilization of FSP1, an anti-ferroptotic protein. The acetylation of FSP1 at the K168 residue is mediated by KAT2B and HDAC3. Inhibiting SLC25A1 and ACLY pharmacologically increases cancer cells' susceptibility to ferroptosis, suggesting that targeting the SLC25A1-ACLY pathway could be an effective strategy to sensitize cells to ferroptosis.

My comments are limited to the molecular mechanism regulating ferroptosis which is my area of expertise. Overall, the study is of interest and provides insights into the biology of FSP1. Nevertheless, the acetylation of FSP1 and its functional implication in regulating ferroptosis appear less convincing. Some aspects of the study could be better explained by alternative mechanisms. Therefore, I would encourage the authors to consider this in order to provide a better mechanistic interpretation.

Major point:

- The connection between SLC25A1 and ACLY in ferroptosis is convincingly demonstrated. Nonetheless an alternative explanation that is independent on the acetylation of FSP1 could explain the sensitization observed. Loss of ACLY is expected to inhibit lipogenesis as the authors rightly acknowledge, can the authors exclude that the loss of ACLY is not promoting a shift in the MUFA/PUFA ratio. The analysis presented in figure EV4 could be more informative if this comparison is presented.

Response: Thank you for this valuable comment. Just for clarification, in our lipidomics analysis in Figure EV4 we knocked down SLC25A1 as it was initially identified in our CRISPR screening of the SLC family. As advised by the reviewer, we have performed further analyses of the lipidomics data and found that SLC25A1 knockdown did not significantly impact the total MUFA/PUFA ratio, nor did it affect the PC-MUFA/PC-PUFA and PE-MUFA/PE-PUFA ratios (Figure A below, **Figure EV4C in the revised manuscript**).

- Following up on my previous point where lack of ACLY could affect the availability of saturated and monounsaturated fatty acids, is there any change in the subcellular distribution of FSP1? If myristic acid, which is required for FSP1 myristoylation, is limiting, then a pool of non-membrane bound FSP1 could result in an increase in ferroptosis sensitivity. If that's the case, one should consider investigating the half-life of wt and non-myristoylated (G2A) FSP1.

Response: This is a very insightful point. We conducted a series of additional experiments to ascertain that FSP1 myristoylation is not implicated in the increased ferroptosis susceptibility induced by the knockdown of SLC25A1 or ACLY.

Myristic acid (MA) is the only donor for the myristoylation of proteins. Firstly, the LC-MS/MS analyses revealed that while levels of palmitic acid (PA) were significantly decreased by silencing of SLC25A1 or ACLY, the levels of MA remained largely unaffected. This discrepancy may be attributed to the relatively low cellular abundance of MA, suggesting that its homeostasis could be more rigorously maintained (Figure A below, **Figure EV4D in the revised manuscript**). We then performed subcellular fractionation to isolate plasma membrane and cytosolic fractions, which demonstrated that knockdown of SLC25A1 or ACLY does not affect the plasma membrane localization of FSP1 (Figure B-C below, **Figure EV7G-7H in the revised manuscript**). Consistently, the immunofluorescence analyses indicated that while knockdown of SLC25A1 or ACLY reduces FSP1 expression, it did not alter the subcellular localization of FSP1 (Figure D below, **Figure EV7I in the revised manuscript**). In addition, we introduced wild-type (WT) FSP1 and the myristoylation-resistant G2A mutant into HEK293T cells. Cycloheximide (CHX) chase assays revealed that the protein half-life of the G2A mutant was comparable to that of WT FSP1 (Figure E below). Collectively, these results indicate that SLC25A1 or ACLY affects FSP1 protein stability and ferroptosis susceptibility independently of FSP1 myristoylation.

- An important experiment the authors might wish to perform in order to provide support for the hypothesis would be to express the non-acetylatable and the acetyl-mimic mutant of FSP1 in FSP1-KO, ACLY/FSP1-2KO and SLC25A1/FSP1-2KO cells and assess their response to RSL3. This should provide unequivocal proof that SLC25A1 and ACLY are increasing ferroptosis sensitivity in an FSP1 dependent manner.

Response: We appreciate the reviewer's suggestion. As recommended, we introduced the acetylation-mimetic K168Q mutant and the non-acetylatable K168R mutant into FSP1-knockout A549 cells with or without depletion of SLC25A1 or ACLY. Our findings indicate that the FSP1 K168Q mutant effectively rescued the increased sensitivity to ferroptosis observed in FSP1-KO cells, while the K168R mutant exhibited a notably weaker protective effect, due to its reduced protein stability (Figure A-B below, **Figure EV8E-8F in the revised manuscript**). Importantly, regardless of the FSP1 mutants, the knockdown of SLC25A1 or ACLY did not further enhance cellular sensitivity to RSL3-induced ferroptosis (Figure A-B below, **Figure EV8E-8F in the revised manuscript**). These results suggest that SLC25A1 and ACLY modulate ferroptosis sensitivity in a FSP1 acetylation-dependent manner.

- Mass spectrometry analysis of immunoprecipitated FSP1 validating its acetylation beyond the use of antibodies its advisable.

Response: We thank the reviewer for raising this issue. As suggested, we immunoprecipitated the Flag-tagged full-length wild-type FSP1 in HEK293T cells and performed LC-MS/MS assay. The MS results indicate that a potential acetylation modification at the K168 residue of FSP1 (Figure A below). Furthermore, the acetylation of FSP1 at the K168 site has been identified in acetylation stoichiometry measurements in human cervical cancer HeLa cells (Nat Commun. 2019 Mar 5;10(1):1055, Supplementary Data 1a).

- The authors use PTGS2 as a read out of "ferroptosis" this is probably a very unspecific approach and should be avoided since multiple stress response pathways can lead to PTGS2 upregulation. I would encourage to validate some of the finding using BODIPY-C11, which is a probe sensitive to alterations in membrane redox state.

Response: We thank the reviewer for raising this issue. We utilized BODIPY-C11 to demonstrate that the knockdown of SLC25A1 or ACLY resulted in increased lipid ROS levels (Figure A-B below, **Figure EV10-1P** and **Figure EV3K-3J** in the revised manuscript). Additionally, as shown in Figures 1K-L and EV1M-N, the ferroptosis inhibitor Fer-1 effectively mitigated the accumulation of RSL3-induced lipid ROS following SLC25A1 knockdown in both A375 and A549 cells as assessed using BODIPY-C11.

Minor point:

- Line 57, PMID: 38297129 needs to be included

Response: We thank the reviewer for pointing out this issue. We have cited this study in our references.

- Correct lipid peroxides to lipid hydroperoxides throughout the text

Response: We have corrected them in the text.

Referee #2:

The work is great. However, some issues remain that should be addressed to better support the concept.

1. Spaces are missing before inserted references in the text.

Response: We thank the reviewer for pointing out this issue. We have adjusted the text accordingly.

2. In figure 1F, efficiency of SLC25A1 overexpression in cells with SLC25A1 knockdown should be shown.

Response: Thanks for the suggestion. We have added the data to Figure 1F.

3. The full name of Fer-1 should be added in line 104 instead of line 111.

Response: We have adjusted the text accordingly.

4. The authors state that SLC25A1 knockdown directly induced cell ferroptosis. But why did MDA staining show no difference between sample with vehicle + sham and sample with Slc25a1 KD in figure 1N.

Response: We thank the reviewer for raising this issue. Malignant cells generally exhibit greater sensitivity to redox fluctuations and ferroptosis induction compared to normal cells or tissues (Science. 2020 Apr 3;368(6486):85-89; Nat Commun. 2019 Apr 8;10(1):1617). This enhanced sensitivity is likely attributed to their highly metabolic state and abnormal mitochondrial function (Exp Mol Med. 2020 Feb;52(2):192-203). Consequently, the knockdown of SLC25A1 led to ferroptosis in cancer cells, primarily due to the downregulation of the FSP1 protein (Figure 3B), as well as a reduction in NADPH and GSH levels (Figure EV3D-E). In contrast, SLC25A1 knockdown alone did not induce significant oxidative stress or ferroptosis in normal liver tissue (Figure 1N).

In addition, the redox microenvironment *in vivo* is different from *in vitro* tumor cell culture. For instance, the serum of mice contains a variety of anti-oxidant metabolites such as vitamin E and vitamin K, which are obtained through diet or gut microbe to defend against ferroptosis. On the contrary, there is very low content of vitamin E and vitamin K in the medium for *in vitro* tumor cell culture. These factors contribute to the scenario that the liver tissues with SLC25A1 KD are more resistant to ferroptosis under normal condition.

5. As the authors focused on liver IRI *in vivo*, why lung cancer cell line A549 was chosen *in vitro* experiments?

Response: Thanks for raising this issue. Our findings indicate that the knockdown of SLC25A1 enhances sensitivity to ferroptosis across various cancer cell lines, including A375, SK-MEL-28 and SK-MEL-30 melanoma cells, HT29 and RKO colorectal cancer cells, as well as A549 lung cancer cells (Figure 1C-D, Figure EV1B-C and F-G). This suggests that the protective role of SLC25A1 against ferroptosis is not tissue-specific. Given that liver ischemia-reperfusion injury (IRI) is associated with ferroptosis (Nat Cell Biol. 2014 Dec;16(12):1180-91), we employed the liver IRI model to illustrate the protective effect of SLC25A1 *in vivo* (Figure 1N). To clarify further, we also silenced SLC25A1 expression in Huh-7 and HepG2 hepatocellular carcinoma cells and observed increased sensitivity to ferroptosis triggered by RSL3 and erastin (Figure A-D below, **Figure EV1E-1G in the revised manuscript**).

6. Noting that figure 2A showed citrate levels in the total cell, cytoplasm, and mitochondria, the protocols for the isolation of cytoplasm and mitochondria should be described in methods. Additionally, fraction contamination should be ruled out by immunoblotting of markers for cytoplasm, nuclear and mitochondria.

Response: Thank you very much for your suggestion. We utilized a commercial isolation kit (Beyotime, C3601) to extract mitochondria, and we compared citrate levels between total cells, mitochondria, and non-mitochondrial fractions (cytosol). Following the reviewer's recommendations, we have included detailed mitochondrial isolation methods in the Methods section. We also performed immunoblotting using the mitochondrial protein TOM20 as a marker for mitochondria and β -actin as a marker for cytosol to rule out any potential contamination (Figure A below, **Figure EV3A in the revised manuscript**).

7. The effect of inhibiting acetyl-CoA production by dimethyl citrate and sodium acetate should be demonstrated to support the result of Figure EV3J-S3K.

Response: We appreciate this suggestion. Our results indicate that treatment with dimethyl citrate or sodium acetate increases acetyl-CoA levels in A375 and A549 cells (Figure A below, **Figure EV3M in the revised manuscript**).

8. The IP result in figure 3H is flawed as the levels of K168R-FLAG in WCL are lower than others.

Response: Thank you for pointing out this issue. Our findings demonstrate that acetylation of FSP1 at K168 enhances its protein stability, as evidenced by the reduced half-life of the acetylation-defective FSP1 K168R mutant (Figure 3J). Consequently, in Figure 3H, despite we introduced the comparable amounts of plasmids expressing different FSP1-Flag mutants, the protein level of K168R-Flag FSP1 in WCL is lower than others.

9. Bands of GAPDH in WCL are missing in Figure 4C, 4K, 5A, 5B, EV6A, EV6B, EV6H.

Response: We have added this data as suggested.

Dear Prof. Yin,

Thank you for submitting the revised version of your manuscript, which addresses the concerns of the referees. This revised version has now been re-reviewed; I attach the second referee reports to the bottom of this mail. As you will see, you have addressed referee #1's concerns. Referee 2 asks for a list of clarifications which I would like you to consider carefully. If you would like to discuss these by Zoom, please let me know. There are also some remaining editorial points which need to be addressed. In this regard, would you please:

- remove the figures from the manuscript file and upload as individual, high-resolution Figure files,
 - resolve the following grant number mis-matches. In manuscript: 2022YFA0912600 / in online submission system: 2022YFA090016; missing in manuscript: GDSTC | Guangdong Provincial Applied Science and Technology Research and Development Program (Guangdong Foundation for Program of Science and Technology Research) 2020A1515110857,
 - remove the author credit section from the manuscript file,
 - use up to 5 EV figures; the others should be compiled in Appendix PDF with the nomenclature Appendix Figure Sx (source file names, titles, legends and ms callouts all need to be Appendix Figure Sx), include a title page with a table of contents and page numbers,
 - include a Reagents and Tools table,
 - provide source data files for Fig. 3J and 7E-F - fill out the source data checklist,
 - provide source data files for Figure 4N and Figure EV6H,
 - ensure datasets HRA007497 and PXD052299 datasets are made publicly available upon acceptance and provide URLs in the data availability statement,
 - rectify the legend for figure EV 1q-s, which is currently mislabeled as figure EV 1o-q,
 - provide exact p values in the legends of figures 1e; 2a-b, f; 3c, e, j; 4e, m; 5e-f; 6c-d, h-k; 7a, c-d; EV 1d, l, n, p-s; EV 2a-b; EV 3b, e-f, k-m; EV 4d, f; EV 5d; Ev 6e, k; Ev 7e-f, EV 8k-m; EV 9b-c,
 - correct mismatch between the annotated p values in the figure legend and the annotated p values in the figure file in figures 1e, j, l-m; 2a-b, f; 3c, e, j; 4e, m; 5e-f; EV 1d, l, n, p-s; EV 2a-b; EV 3b, e-f, k-m; EV 4c-d, f; EV 5d; EV 6e, k; EV 7e-f,
 - define n in the legends of figures 1b; 7a, c-f,
 - although 'n' is provided, describe the nature of entity for 'n' in the legends of figures 1c-e, g-h, j, l-m; 2a-b, d-f, h-i; 3c, e, j, l-m; 4e, m, o-p; 5e-f, i-j, m-n; 6c-d, k; EV 1b-d, f-g, i-j, l, n, p, q-s; EV 3b-f, h-l, k-o; EV 4c-d, f; EV 5d, f; EV 6e, k; EV 7e-h; EV 8c-d, f, i-m,
 - label axis gaps appropriately in figure EV 2a, and
 - rename tables EV1-EV3 as Table 1-3 with corresponding callouts,
- correct the section order as following: title page with complete author information, abstract, keywords, introduction, results, discussion, methods, data availability section, acknowledgements, disclosure and competing interests statement, references, main figure legends, tables, expanded figure legends.

We include a synopsis of the paper (see <http://emboj.embopress.org/>). Please provide me with a general summary image, two-sentence summary statement and 3-5 bullet points that capture the key findings of the paper.

I look forward to receiving these changes. EMBO Press is an editorially independent publishing platform for the development of EMBO scientific publications.

Best wishes,

William

William Teale, PhD
Editor
The EMBO Journal
w.teale@embojournal.org

We realize that it is difficult to revise to a specific deadline. In the interest of protecting the conceptual advance provided by the work, we recommend a revision within 3 months (6th Feb 2025). Please discuss the revision progress ahead of this time with the editor if you require more time to complete the revisions. Use the link below to submit your revision:

Referee #1:

The authors have diligently worked to address my comments. They have significantly strengthened their conclusion with a large set of new experiments. Based on the new data presented, I support the publication of the manuscript.

Referee #2:

The authors have done a lot of additional great work to further support their conclusion. But the connection between ACLY and SLC25A1 wasn't elucidated clearly.

1. In Fig EV3A, was the amount of whole cell lysates equal to mitochondrial fraction and cytosolic proteins? If equal, levels of mitochondrial SLC25A1 and TOM20 should be higher than those WCL. In addition, IB of nuclear biomarker such as lamin B1 should also be performed to exclude nuclear contamination.

2. The inhibitory effect of Fer-1 in Fig EV3L is too weak to support further research though it showed statistical difference.

3. What's the y-axis title of Fig EV5G? Why such result could reflect enzymatic activity of FSP1?

4. The SLC25A1-ACLY-FSP1 acetylation axis wasn't convincingly validated as the following dictated.

(1) Besides citrate, acetyl-CoA could also be converted from acetate by ACSS1/2 or from pyruvate by PDC. Fig EV3G showed that SLC25A1 knockdown upregulated pyruvate metabolism which was reported to enhance ferroptosis. The authors should exclude effect of pyruvate metabolism on ferroptosis to better elucidate the role of FSP1 acetylation by SLC25A1. In addition, conversion of acetyl-CoA from acetate happens not only in mitochondria and cytoplasm, but also in nucleus where SLC25A1 doesn't exist. (2) ACLY is the key molecule in lipogenesis and its inhibition could promote PUFA peroxidation and mitochondrial damage. Though the authors proved that SLC25A1 doesn't affect PUFA/MUFA ratio, they didn't exclude role of ACLY-PUFA axis in ferroptosis.

Point-by-point response to the reviewers' comments

Referee #1

The authors have diligently worked to address my comments. They have significantly strengthened their conclusion with a large set of new experiments. Based on the new data presented, I support the publication of the manuscript.

Response: Thanks for the positive feedback and support for the publication of our manuscript.

Referee #2:

The authors have done a lot of additional great work to further support their conclusion. But the connection between ACLY and SLC25A1 wasn't elucidated clearly.

1. In Fig EV3A, was the amount of whole cell lysates equal to mitochondrial fraction and cytosolic proteins? If equal, levels of mitochondrial SLC25A1 and TOM20 should be higher than those WCL. In addition, IB of nuclear biomarker such as laminB1 should also be performed to exclude nuclear contamination.

Response: Thank the reviewer for raising the insightful points. In Fig. EV3A, the amount of whole-cell lysates (WCL) was equal to that of the cytosolic fraction but was not matched to the mitochondrial fraction. The mitochondrial fraction was isolated using a commercial kit optimized for high purity, which, while effective, can lead to some loss of mitochondrial proteins and relatively low protein concentrations. Consequently, our initial focus was on confirming the purity of isolated mitochondrial rather than achieving direct comparison with WCL.

Additionally, due to the similar molecular weights of SLC25A1 and TOM20, we separated their detection by running them on distinct SDS-PAGE gels. To ensure clear and distinct bands, we optimized sample loading based on the sensitivities of the respective antibodies. Since the SLC25A1 antibody demonstrates greater sensitivity compared to the TOM20 antibody, we loaded more protein for TOM20 detection. This adjustment accounts for the observation that mitochondrial SLC25A1 and TOM20 levels are not uniformly higher than those in the WCL.

In response to the reviewer's valuable suggestions, we have repeated the experiment with equal protein loading across fractions. Furthermore, we conducted immunoblotting for the nuclear marker Lamin B1 to evaluate potential nuclear contamination in the mitochondrial

fraction. For accuracy, we have also rephrased “cytosol” in the previous version to “cytosol and nucleus”. The updated data are now included in **Appendix Fig. S3A** of the revised manuscript (Figure A below).

2. The inhibitory effect of Fer-1 in Fig EV3L is too weak to support further research though it showed statistical difference.

Response: We thank the reviewer for the valuable comment. ACLY inhibition has been reported to suppress cancer cell proliferation and induce apoptosis (Cancer Discov. 2019 Mar;9(3):416-435). In our study, ACLY knockdown alone significantly increased lipid ROS levels (**Appendix Fig. S3J-K**), suggesting a potential direct role of ACLY in inducing ferroptosis in cancer cells. To investigate this, we treated ACLY-depleted A375 and A549 cells with Fer-1 (ferroptosis inhibitor), Z-VAD-fmk (Z-V, apoptosis inhibitor), Necrostatin-1 (Nec-1, necrosis inhibitor), and Bafilomycin A1 (Baf-A1, autophagy inhibitor). Both Z-V and Fer-1 significantly, though not completely, mitigated ACLY silencing-induced cell death (**Appendix Fig. S3L**), indicating that ACLY knockdown triggers both apoptosis and ferroptosis.

While the inhibitory effect of Fer-1 in **Appendix Fig. S3L** appears moderate under basal conditions, it is important to highlight that in the presence of the ferroptosis inducer RSL-3, where ferroptosis predominates, ACLY depletion markedly exacerbated cell death (**Fig. 2F**). This effect was almost entirely rescued by Fer-1, strongly supporting the conclusion that ACLY knockdown substantially enhances ferroptosis sensitivity.

3. What's the y-axis title of Fig EV5G? Why such result could reflect enzymatic activity of FSP1?

Response: Thank you for pointing out the missing y-axis title in Fig. EV5G, and we apologize for this oversight. The y-axis in Fig EV5G represents the levels of oxi-STY-BODIPY, which efficiently reflects the level of lipid peroxidation in vitro. We measured the levels of oxi-STY-BODIPY using the fluorescence-enabled inhibited autoxidation (FENIX) assay. This assay was conducted following established protocols (Nature, 2019 Oct 21;575:693–698; Nature, 2022 Aug 3;608:778–783; Nat. Chem. Biol., 2020 Aug 10;16:1351–1360).

FENIX is based on the kinetic competition between radical-trapping antioxidants (RTAs) and phospholipids for peroxy radicals, where RTAs yield a persistent radical intermediate (Cell Chem. Biol., 2019 Nov 21;26:1594–1607.e7). STY-BODIPY serves as the fluorescent probe, whose fluorescence increases significantly upon reaction with peroxy radicals. The assay incorporates egg phosphatidylcholine (PC) to mimic cellular membranes, di-tert-decyl hyponitrite (DTUN) as the radical initiator, and vitamin K (VK) as the substrate. FSP1 uses NADPH to generate reduced VK, which acts as the RTA to diminish peroxy radicals (Nature, 2022 Aug 3;608:778–783). Therefore, the enzymatic activity of FSP1 can be reflected by the slope of the fluorescence increase over time. A steeper slope indicates lower enzymatic activity. Liprostatin-1 (Lip-1) was used as a positive control in this experiment. The results demonstrate the relationship between FSP1 activity and the kinetic competition for peroxy radicals. The updated data are now included in **Appendix Fig. S6G** of the revised manuscript (Figure A below).

4. The SLC25A1-ACLY-FSP1 acetylation axis wasn't convincingly validated as the following dictated.

(1) Besides citrate, acetyl-CoA could also be converted from acetate by ACSS1/2 or from pyruvate by PDC. Fig EV3G showed that SLC25A1 knockdown upregulated pyruvate metabolism which was reported to enhance ferroptosis. The authors should exclude effect of pyruvate metabolism on ferroptosis to better elucidate the role of FSP1 acetylation by SLC25A1. In addition, conversion of acetyl-CoA from acetate happens not only in mitochondria and cytoplasm, but also in nucleus where SLC25A1 doesn't exist.

Response: Thank the reviewer for raising these important points. We acknowledge that acetyl-CoA can be derived from multiple sources, including acetate (via ACSS1/2) and pyruvate (via PDC), and that acetyl-CoA is present in mitochondria, cytoplasm, and nucleus. To better elucidate the specific role of SLC25A1 in regulating FSP1 acetylation and ferroptosis, we conducted additional experiments to assess the contributions of ACSS1/2 and PDC.

To determine whether ACSS1/2 modulates ferroptosis sensitivity or FSP1 acetylation, we knocked down ACSS1 and ACSS2 in A375 cells. Neither knockdown affected sensitivity to RSL3-induced ferroptosis (Figure A-D below, **Appendix Fig. S4A-B and S4D-E in the revised manuscript**) nor altered FSP1 acetylation levels (Figure E-F below). Additionally, in SLC25A1- and ACLY-depleted A375 cells, silencing ACSS1/2 did not affect the enhanced ferroptosis sensitivity induced by SLC25A1 or ACLY depletion (Figure G-N below, **Appendix Fig. S4G-H, S4J-K, S4M-N and S4P-Q in the revised manuscript**). Interestingly, supplementation with sodium acetate partially alleviated the increased ferroptosis observed in SLC25A1- or ACLY-depleted cells (Appendix Fig. 4T-U and Figure O-P below, **Appendix Fig. S4V-W in the revised manuscript**) and restored the reduced FSP1 acetylation levels (Figure Q-R below, **Appendix Fig. S8C-D in the revised manuscript**). These findings suggest that ACSS1/2 plays a secondary role in acetyl-CoA production for FSP1 acetylation. This is consistent with previous reports indicating that ACSS-dependent acetyl-CoA production is primarily utilized under metabolic stress conditions, such as hypoxia, lipid depletion or ACLY inhibition (Cancer Cell. 2015 Jan 12;27(1):57-71; Cell Rep. 2016 Oct 18;17(4):1037-1052). In the absence of exogenous acetate supplementation, SLC25A1 remains the primary regulator of ferroptosis sensitivity.

Our KEGG pathway analysis of proteomics data showed that SLC25A1 knockdown upregulates pyruvate metabolism (Appendix Fig. 5G). It has been reported that knockdown of the E1 subunit of the pyruvate dehydrogenase complex (PDHA1) in HT1080 cells, under high pyruvate conditions (10 mM), protects against erastin-induced ferroptosis (Free Radic Biol Med, 2021, 167:45-53). To explore the role of PDHA1 in FSP1 acetylation and ferroptosis sensitivity, we knocked down PDHA1 in A375 cells. However, its depletion did not affect sensitivity to RSL3-induced ferroptosis (Figure S-T below, **Appendix Fig. S4C and S4F in the revised manuscript**) or alter FSP1 acetylation levels (Figure U below). Moreover, PDHA1 knockdown did not influence the enhanced ferroptosis sensitivity observed upon SLC25A1 or ACLY depletion (Figure V-Y below, **Appendix Fig. S4I, S4O, S4L and S4R in the revised manuscript**). These results indicate that SLC25A1/ACLY-mediated FSP1 acetylation and ferroptosis sensitivity are independent of the PDC-mediated pyruvate metabolism pathway.

As the reviewer pointed out, acetyl-CoA is also present in the nucleus, where it is primarily used for histone acetylation and transcriptional regulation (Nat Rev Cancer, 2023, 23(3):156-172). However, as shown by our RNA sequencing data, the regulatory role of SLC25A1 in ferroptosis sensitivity is not at the transcriptional level (Appendix Fig. S5F). Furthermore, FSP1 is predominantly localized in the cytosol and plasma membrane, making it unlikely that nuclear acetyl-CoA directly affects its acetylation. Therefore, nuclear acetyl-CoA does not affect the regulatory role of SLC25A1 in ferroptosis sensitivity or FSP1 acetylation.

(2) ACLY is the key molecule in lipogenesis and its inhibition could promote PUFA peroxidation and mitochondrial damage. Though the authors proved that SLC25A1 doesn't affect PUFA/MUFA ratio, they didn't exclude role of ACLY-PUFA axis in ferroptosis.

Response: Thanks for raising this issue. We agree that ACLY plays a key role in lipogenesis and that its inhibition can promote PUFA peroxidation, potentially influencing ferroptosis sensitivity. Indeed, cancer cells often maintain higher levels of de novo fatty acid synthesis to avoid excess PUFA uptake, which can lead to lipid peroxidation and increased oxidative stress sensitivity (Cancer Res. 2010 Oct 15;70(20):8117-26). As ACLY inhibition impairs de novo fatty acid synthesis, it increases the demand for PUFAs, leading to their accumulation and subsequent peroxidation (Sci Adv, 2023; 9(49): eadi2465; Sci Adv, 2023; 9(18): eadf0138).

However, as shown in Appendix Fig. S9E-F, in FSP1-knockout cells, as well as in FSP1-K168R or FSP1-K168R cells, ACLY inhibition did not further enhance sensitivity to RSL3-induced ferroptosis. Conversely, overexpression of ACLY in FSP1-knockout cells did not reduce ferroptosis sensitivity (Figures 5N, Appendix Fig. S9J). In contrast, overexpression of FSP1 significantly mitigated the sensitivity of ACLY-depleted A375 and A549 cells to RSL3- and erastin-induced ferroptosis (Figures 5J, Appendix Fig. S9D). These findings suggest that the regulation of ferroptosis sensitivity by ACLY occurs largely through FSP1 acetylation. Therefore, while ACLY plays a role in PUFA metabolism, its modulation of ferroptosis sensitivity is dependent on FSP1, rather than directly through the ACLY-PUFA axis alone.

Dear Prof. Yin,

I am pleased to inform you that your manuscript has been accepted for publication in the EMBO Journal.

Congratulations!

Yours sincerely,

William Teale

William Teale, PhD
Editor
The EMBO Journal
w.teale@embojournal.org
